# Rethinking Fourier Transform from A Basis Functions Perspective for Long-term Time Series Forecasting

**Runze Yang[1,2], Longbing Cao[2*], Jianxun Li[1], Jie Yang[1*]**
[1]Department of Automation, Shanghai Jiao Tong University
[2]School of Computing, Macquarie University
{runze.y, jieyang, lijx}@sjtu.edu.cn
{runze.yang}@hdr.mq.edu.au, {longbing.cao}@mq.edu.au

## Abstract

The interaction between Fourier transform and deep learning opens new avenues for long-term time series forecasting (LTSF). We propose a new perspective to reconsider the Fourier transform from a basis functions perspective. Specifically, the real and imaginary parts of the frequency components can be viewed as the coefficients of cosine and sine basis functions at tiered frequency levels, respectively. We argue existing Fourier-based methods do not involve basis functions thus fail to interpret frequency coefficients precisely and consider the time-frequency relationship sufficiently, leading to inconsistent starting cycles and inconsistent series length issues. Accordingly, a novel Fourier basis mapping (FBM) method addresses these issues by mixing time and frequency domain features through Fourier basis expansion. Differing from existing approaches, FBM (i) embeds the discrete Fourier transform with basis functions, and then (ii) can enable plug-and-play in various types of neural networks for better performance. FBM extracts explicit frequency features while preserving temporal characteristics, enabling the mapping network to capture the time-frequency relationships. By incorporating our unique time-frequency features, the FBM variants can enhance any type of networks like linear, multilayer-perceptron-based, transformer-based, and Fourier-based networks, achieving state-of-the-art LTSF results on diverse real-world datasets with just one or three fully connected layers. The code is available at: https://github.com/runze1223/Fourier-Basis-Mapping.

## 1 Introduction

Long-term time series forecasting (LTSF) is essential for applications with long-range sequences, which presents significant challenges including long-range temporal dependencies, and frequency-oriented global dynamics. Recently, deep neural networks (DNNs) have thrived to tackle LTSF challenges for the presence of hierarchical effects, varied outliers, and nonlinear dynamics. They use various DNN architectures including recurrent neural networks (RNNs) [1, 2, 3, 4, 5, 6, 7, 8], convolution neural networks (CNNs) [9, 10, 11, 12, 13, 14], multi-layer perceptron (MLP) based networks [15, 16, 17, 18], Transformer-based networks [19, 20, 21, 22, 23, 24, 25, 26, 27, 28, 29, 30, 31, 32, 33]. However, a recent study NLinear [34] shows that an embarrassingly simple normalized linear model can outperform most of the DNN-based methods. This raises the question: *Would a complex Transformer-like architecture necessarily lead to better LTSF performance?* According to CrossGNN [35], DNN-based methods like Transformers suffer from the influence of noise signals and produce high attention scores for unexpected noises.

---

[*]Corresponding author

38th Conference on Neural Information Processing Systems (NeurIPS 2024).

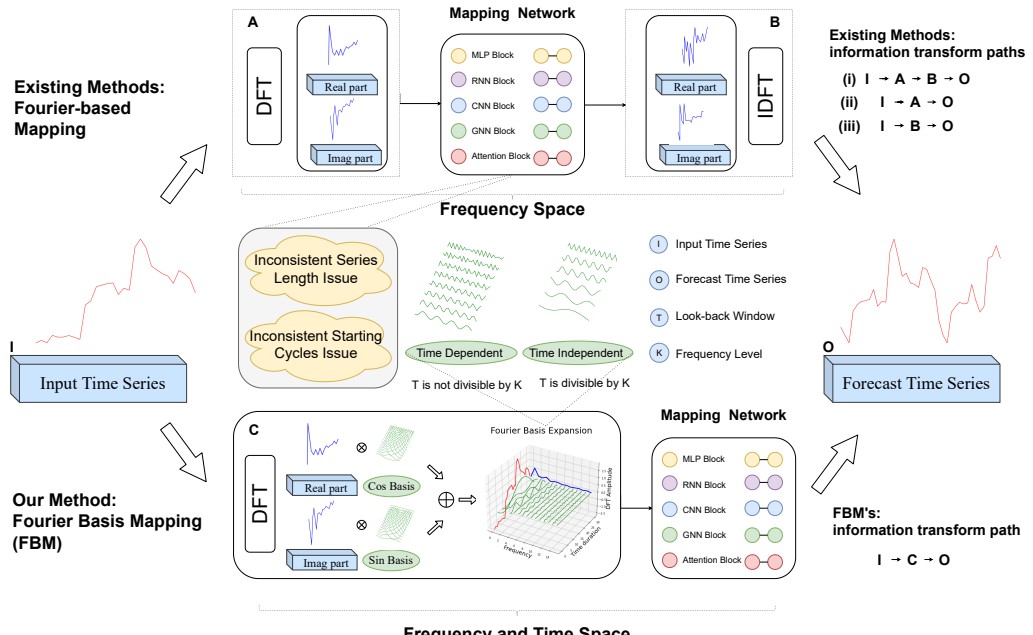

Figure 1: Comparison of Existing Fourier-based Methods with Our Approach Fourier basis mapping. Existing methods primarily operate in the frequency domain, with the mapping focusing on frequency features. Our FBM simultaneously operates in both the time and frequency domains, with the mapping focusing on time-frequency features.

Thus, Fourier-based time series modeling emerges as a new paradigm to remove noise signals by considering diverse effects hierarchically at different frequency levels. If we rethink the Fourier transform from a basis functions perspective, the real and imaginary parts can be interpreted as the coefficients of cosine and sine basis functions at tiered frequencies, respectively. However, existing Fourier-based methods do not involve basis functions, thus failing to interpret frequency coefficients precisely and do not consider the time-frequency relationships sufficiently, as shown in Figure 1. They face two main issues: inconsistent starting cycles and inconsistent series length issues. For instance, FEDformer [22], FreTS [18], FiLM [36], FITS [37], FGNet [28], and FL-Net [38] use the real and imaginary parts of frequency components as the input of the mapping network and conduct the mapping in the frequency space. From our new perspective, we find that the amplitude and arctangent of the real and imaginary components carry more explicit meanings than the components themselves. Furthermore, it is worth noting that the meanings of frequency components are bounded by the series length, and failing to establish the connections will lead to ambiguity in understanding the precise frequency in these models, e.g., 2 Hz sine and cosine wave with series lengths 96 and 336 has distinct meaning. More importantly, Figure 1 shows that a Fourier basis function is time-dependent when the input length is not divisible by a certain frequency level, making it even more challenging for the model to correctly interpret those frequency components without the basis functions. Consequently, constructing the mapping in the frequency space is not enough and cannot capture time-frequency relationships, which has been ignored by all existing Fourier-based methods. Although some methods like TimesNet [39] introduce time-frequency features, these features are non-interpretable and complex, as the summation over the frequency dimension does not recover the original time series. We provide a more detailed discussion of their limitations in Section 2.

Accordingly, we propose the Fourier Basis Mapping (FBM) to address the aforementioned issues, involving learnable and non-learnable features. In the first stage, we embed the discrete Fourier transform with basis functions, referred to as *Fourier basis expansion*, to extract time-frequency features. In the second stage, we validate its effectiveness using three types of mapping networks through three FBM variants (FBM-L, FBM-NL, FBM-NP) against four categories of LTSF baseline methods: (1) Linear method: NLinear; (2) Transformer-based methods: iTransformer [33] and PatchTST [20]; (3) MLP-based methods: N-BEATS [16], FreTS, and TimeMixer [15]; and (4) Fourier-based methods: N-BEATS, FITS, FreTS, and CrossGNN. The experimental results reveal

that Fourier basis expansion can enhance any types of DNNs. With time-frequency features, simple mapping networks (e.g, L-vanilla linear networks and NL-three-layer MLPs) can achieve the SOTA of LTSF on eight real-world datasets and of STSF on PEMS datasets.

## 2 Related work

Recent studies have investigated the effect of the Fourier transform in addressing LTSF challenges. Consequently, we categorize the relevant studies into two groups: (1) Fourier-based architectures, and (2) DNN-based architectures. In the first category, we discuss the applications of the Fourier transform, while the second addresses innovative model architectures.

### 2.1 Fourier-based architectures

Fourier transform has been integrated into a wide range of network architectures, including CNNs [39], recurrent networks [36], graph neural networks (GNNs) [35, 40], Transformer [22], and MLP networks [18, 16, 17]. However, by rethinking the Fourier transform from a basis functions perspective, we identify the inconsistent starting cycles and series length issues. In path (i) of Figure 1, methods like FEDformer [22], FreTS [18], FiLM [36], FITS [37], FGNet [28], and FL-Net [38] use real and imaginary parts as inputs to their mapping networks but the networks cannot easily interpret those coefficients because the crucial information is stored in the amplitude, phase and length of each cycle. In path (ii) of Figure 1, methods like CrossGNN [35], and TimesNet [39] use the discrete Fourier transform (DFT) to select the top-k amplitudes for noise filtering. However, a higher amplitude does not necessarily indicate a useful frequency and a lower amplitude is not necessarily useless. In addition, CrossGNN only stores the values of the first cycles without considering the effects of phase shifts and time-dependent basis, while TimesNet introduces the multi-window Fourier transform, which compromises the integrity of the temporal information. In path (iii) of Figure 1, methods like N-BEATS [16] and N-Hits [17] can similarly be viewed as forecasting through the inverse discrete Fourier transform (IDFT) by computing the output frequency spectrum of the time series, but their networks fail to capture time-dependent effects. In contrast, our method distinguishes itself from existing approaches by leveraging the Fourier basis expansion to provide a mixture of time and frequency domain features, thus avoiding the aforementioned issues.

### 2.2 DNN-based architectures

Firstly, we introduce key techniques for innovative model architectures. In [41], initial normalization is introduced to enhance the robustness. Autoformer [24] advocates the trend and seasonal decomposition with the moving average kernel, consistent with other methods like DLinear [34] and TimeMixer [15]. However, the effectiveness of the decomposition relies on the choice of kernel size, which is improved by our methods. Methods like VH-NBEATS [42] and BasisFormer [21] involve the pre-trained basis functions to account for hourly, daily, and weekly effects. In contrast, our method uses the Fourier basis to capture those effects automatically. Transformer-based methods are one of the most popular LTSF architectures. Some transformer variants improve efficiency by reducing computational complexity and memory cost, such as LogTrans [43], Pyraformer [25], Informer [23], and Autoformer [24]. On the other hand, PatchTST [20] introduces patching to treat a period of local time steps as a semantic vector, effectively reducing complexity and achieving state-of-the-art results over Transformer-based methods. In a recent study, iTransformer [33] suggests that employing attention layers to capture relationships between variables may yield better results. In this paper, we validate the effectiveness of Fourier basis expansion on linear, MLP, and Transformer-based networks, proving that it can enhance any types of networks. The Fourier basis expansion can also be viewed as an enhanced version of trend and seasonal decomposition, as the basis functions, shown in Figure 1, automatically decompose various effects by frequency levels.

## 3 Rethinking Fourier transform w.r.t. basis functions

In this section, we offer a new perspective of the Fourier Transform for LTSF. Firstly, we discuss the mathematical reasoning behind the DFT and IDFT from the perspective of basis functions. From this new perceptive, we find that the real and imaginary parts of the frequency components can be viewed as the coefficients of cosine and sine basis functions at tiered frequency levels. Consequently,

the effect of the IDFT for frequency-based mapping can be seen as analogous to the design of the N-BEATS seasonal block. As such, we identify the inconsistent starting cycles and inconsistent series length issues in existing studies.

Let $\mathbf{X}$ and $\mathbf{Y}$ be the input and output time series, respectively, and $T$ and $L$ represent the look-back window and forecast horizon, assuming both of them are even numbers. $\mathbf{H}^X$ and $\mathbf{H}^Y$ refer to the frequency spectrum of the input and the output, respectively. Then, DFT and IDFT of the input time series $\mathbf{X}$ can be expressed as follows:

$$\mathbf{H}(k) = DFT(\mathbf{X}) = \sum_{n=0}^{T-1} \mathbf{X}[n] \exp\left(-i\frac{2\pi kn}{T}\right), \quad k = 0, 1, \ldots, T-1, \tag{1}$$

$$\mathbf{X}[n] = IDFT(\mathbf{H}) = \frac{1}{T}\sum_{k=0}^{T-1} \mathbf{H}[k] \exp\left(i\frac{2\pi kn}{T}\right), \quad n = 0, 1, \ldots, T-1. \tag{2}$$

From the perspective of basis functions, both DFT and IDFT can be expressed by $\frac{T}{2} + 1$ orthogonal cosine basis functions and $\frac{T}{2} - 1$ orthogonal sine basis functions. This is because the DFT of a real-valued signal is Hermitian symmetric. Their proofs and further explanations can be found in Appendices B.1 and B.2. Subsequently, we can rewrite the IDFT w.r.t. basis functions, and the connection between $\mathbf{X}$ and $\mathbf{H}^X$ can be expressed as follows:

$$\mathbf{X}[n] = \frac{1}{T}\sum_{k=0}^{\frac{T}{2}} \left(\mathbf{a_k} \cos\left(\frac{2\pi kn}{T}\right) - \mathbf{b_k} \sin\left(\frac{2\pi kn}{T}\right)\right), \quad n = 0, 1, \ldots T-1,$$

$$\mathbf{a_k} = \begin{cases} \mathbf{H_R}[k], & k = 0, \frac{T}{2} \\ 2 \cdot \mathbf{H_R}[k], & k = 1, \ldots, \frac{T}{2} - 1 \end{cases} \quad \mathbf{b_k} = \begin{cases} \mathbf{H_I}[k] = 0, & k = 0, \frac{T}{2} \\ 2 \cdot \mathbf{H_I}[k], & k = 1, \ldots, \frac{T}{2} - 1. \end{cases} \tag{3}$$

$\mathbf{H_R}[k]$ and $\mathbf{H_I}[k]$ represent the real and imaginary parts of $\mathbf{H}[k]$ respectively, where $k$ refers to the frequency level, and $\mathbf{H}[k] = \mathbf{H_R}[k] + i\mathbf{H_I}[k]$. Eq. (4) provides an essential insight that the real and imaginary parts of the frequency spectrum can be interpreted as the coefficients of cosine and sine basis functions, respectively. Hence, computing the frequency spectrum of the output time series is equivalent to computing the coefficients between cosine and sine basis functions, which can be considered analogous to the N-BEATS-seasonal-block design. Further explanation and proofs can be found in Appendix B.3.

Consequently, we identify two issues in existing Fourier-based studies: inconsistent starting cycles and inconsistent series length issues. The inconsistent starting cycles issue arises since the real and imaginary parts do not carry explicit meanings without the basis functions. This is because that adding a sine wave and a cosine wave with the same frequency results in a shifted cosine wave with the same frequency, which can be expressed as follows:

$$\mathbf{Z}(t) = A\cos(wt) + B\sin(wt) = R\cos(wt - \phi),$$
$$R = \sqrt{A^2 + B^2}, \quad \phi = arctan(B, A). \tag{4}$$

Therefore, the key information is embedded in the amplitude and arctangent of the real and imaginary values rather than the values themselves. The inconsistent series length issue arises since the meaning of frequency in hertz (Hz) is bound by the series length.

Figure 2 illustrates these two issues as Cases I and II through a manually generated time series with pure sine and cosine basis functions, assuming using time series $\mathbf{X}$ to forecast time series $\mathbf{Y}$. In Case I, $\mathbf{X}$ and $\mathbf{Y}$ have the same frequency and series length but different starting cycles. If the real and imaginary parts of the frequency spectrum of the input are used to compute that of the output through independent channels, we cannot find such a mapping as there is no mathematical solution. However, their relationships can be easily identified through the amplitude and arctangent of the real and imaginary values, which are embedded in basis functions. As shown in Figure 2, a mathematical solution exists which is achievable through Eq. (4).

In Case II, $\mathbf{X}$ and $\mathbf{Y}$ have almost the same frequency and starting cycles but different series lengths. Although the landscapes of the frequency spectrum look similar, their components (in Hz) have

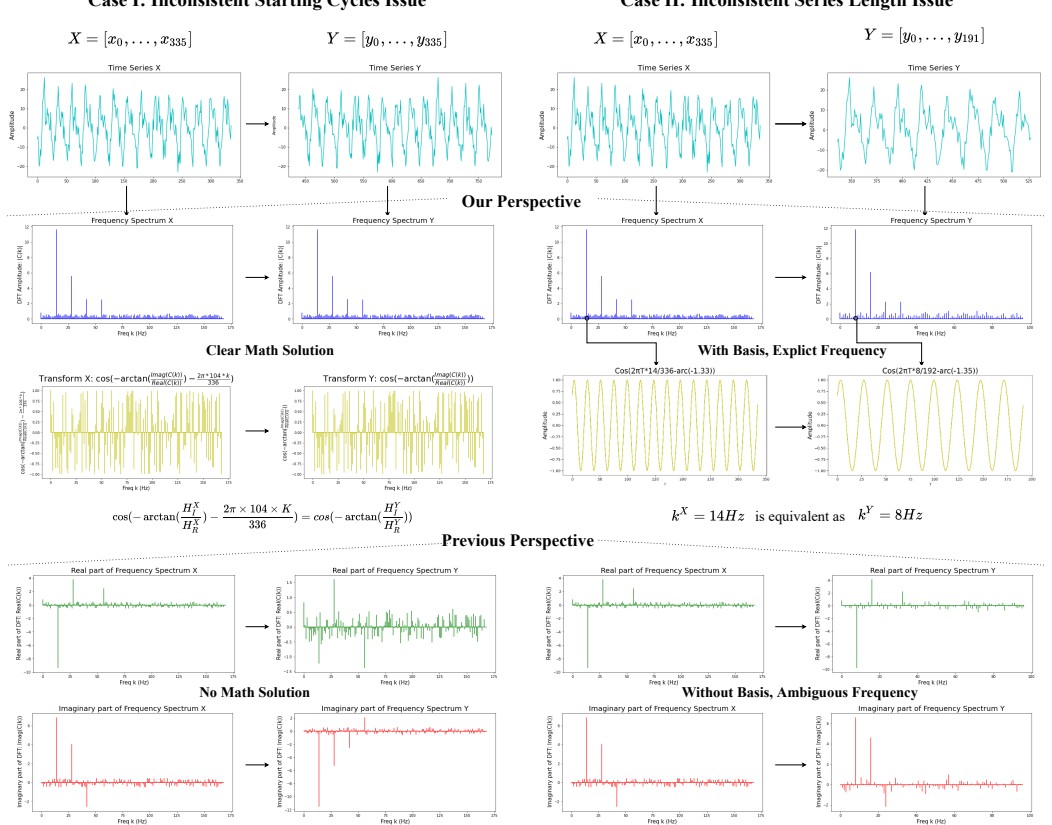

Figure 2: Two Issues of Existing Fourier-based LTSF Models: Inconsistent Starting Cycles Issue and Inconsistent Series Length Issue. In Case I, $\mathbf{X}$ and $\mathbf{Y}$ have a starting cycle gap of 104 over 336. In Case II, $\mathbf{X}$ and $\mathbf{Y}$ have different series lengths of 336 and 192, respectively.

distinct meanings. For instance, an 8Hz cosine function with a series length of 192 is equivalent to a 14Hz cosine function with a series length of 336. Unfortunately, such a frequency connection is not provided by existing models. Hence, the precise interpretation of frequency components is challenging for the model due to varying series lengths. The presence of time-dependent basis functions makes it even harder for the model to accurately interpret the frequency components. Instead, these issues can be solved easily by the incorporation of basis functions, which will be discussed in Section 4.

## 4 FBM: Fourier basis mapping

To address two aforementioned issues, we introduce the Fourier basis mapping (FBM) . Figure 3 shows the architecture of FBM. The primary strength of FBM lies in constructing time-frequency features, which capture explicit frequency information while preserving temporal characteristics. Subsequently, the downstream mapping considers the time-frequency space rather than solely the time or frequency space for forecasting.

First, we apply data normalization to enhance the robustness of FBM, the same as the implementation suggested in [41]. Then, we generate the frequency spectrum $H$ using DFT. Next, we multiply the real part of $\mathbf{H}$ (denoted as $\mathbf{H_R}$) with the orthogonal cosine basis $\mathbf{C}$ and the imaginary part of $\mathbf{H}$ (denoted as $\mathbf{H_I}$) with the orthogonal sine basis $\mathbf{S}$ to obtain the mixture of frequency and temporal features. Let $\mathbf{N} = [0, 1, \ldots, T-1]$, then $\mathbf{C}$ and $\mathbf{S}$ can be expressed as follows:

$$
\begin{aligned}
\mathbf{C} &= \frac{1}{T}[\mathbf{1}, 2\cos(\frac{2\pi\mathbf{N}}{T}), \ldots, 2\cos(\frac{(T-1)\pi\mathbf{N}}{T}), \cos(\pi\mathbf{N})], \\
\mathbf{S} &= -\frac{1}{T}[0, 2\sin(\frac{2\pi\mathbf{N}}{T}), \ldots, 2\sin(\frac{(T-1)\pi\mathbf{N}}{T}), \sin(\pi\mathbf{N})].
\end{aligned}
\tag{5}
$$

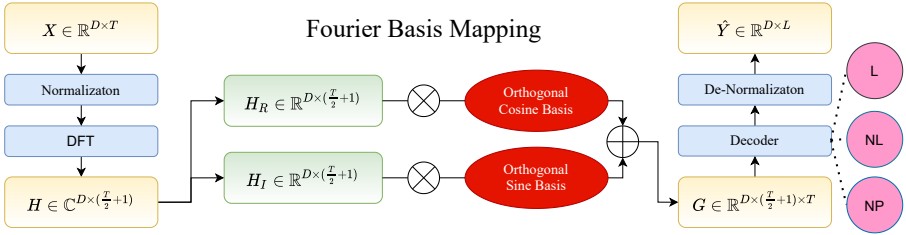

Figure 3: Architecture of Fourier Basis Mapping (FBM): with three feature-output mapping methods, denoted as L - vanilla linear network, NL - nonlinear MLP, and NP - nonlinear PatchTST.

Further, we combine the cosine and sine basis functions to obtain the shifted starting cycles of the time series, forming $\mathbf{G}$. The scalar 2 is derived in Eq. (4) for Hermitian symmetric. Thus, by summing over the $\frac{T}{2} + 1$ frequency domain, the model can recover the original time series without compromising the integrity of the temporal information. The process is analogous to decomposing the original time series into $\frac{T}{2} + 1$ components with tiered frequency levels.

Finally, we use a decoder to map the features $\mathbf{G}$ to the output time series. Our time-frequency features automatically decompose hierarchical effects by frequency levels, accounting for their interactive effects while filtering out associated noise through the mapping network. The Fourier basis expansion provides more effective and interpretable initial features, facilitating a simpler downstream mapping process. To this end, we further propose three mapping methods: linear (L), nonlinear three-layer perception (NL), and nonlinear PatchTST (NP). Consequently, three FBM variants are generated: FBM-L with linear network, FBM-NL with MLP network, and FBM-NP with transformer-based network with patching. The first one includes a single vanilla linear layer, the second one consists of three fully connected layers with activation functions, and the last one follows the same structure as PatchTST, but we conduct the patching after the Fourier basis expansion. It is worth noting that FBM-NP employs fewer patches than PacthTST to reduce complexity, making it more efficient while delivering superior performance. Details of our three mapping networks, including hyperparameters and layer specifications, are provided in Figure 6 and Table 5 in Appendix C.

## 5 Experiment results

### 5.1 Baselines and experiment settings

We compare FBM with eight baseline methods: NLinear, PatchTST, N-BEATS, CrossGNN, FED-former, FreTS, FiLM, and BasisFormer. These methods are chosen because they represent four categories of LTSF modeling methods: (1) Linear method: NLinear; (2) Transformer-based methods: iTransformer and PatchTST; (3) MLP-based methods: N-BEATS, FreTS, and TimeMixer; and (4) Fourier-based methods: N-BEATS, FITS, FreTS, and CrossGNN. Furthermore, FBM is evaluated w.r.t. its three variants: FBM-L, FBM-NL, and FBM-NP. The mean squared error (MSE) and mean absolute error (MAE) are used as evaluation metrics. The look-back window is set to 336, and the forecasting horizons are set to $96, 192, 336, 720$. In Appendix A.3, we also show that a look-back window of 336 is always associated with better results than that of 96 for most baseline methods. Eight datasets are used for the evaluation: ETTh1, ETTh2, ETTm1, ETTm2, Electricity, Traffic, Weather, and Exchange. In addition, we also evaluate the STSF of four PEMS datasets: PEMS03, PEMS04, PEMS07, PEMS08. More details about these baselines and datasets can be found in Appendix A.1. We split the ETT dataset into 12/4/4 months and the other datasets into training, validation, and test sets by the ratio of 6.5/1.5/2. Here, we use a smaller training set and a larger validation set compared to baseline methods to prevent early stopping and ensure a fairer comparison. With time-frequency features, the model can achieve state-of-the-art performance for both LTSF and STSF most of the time using either one linear layer or three non-linear layers.

### 5.2 Effects of FBM mechanisms for LTSF

We evaluate the forecasting performance of three FBM variants against eight diversified baselines incorporated by different architectures: Linear, MLP, Transformer, and Fourier-based architectures. Table 1 shows the quantitative results on eight datasets. While preserving a simple structure, FBM variants achieve the best performance on most datasets across most prediction horizons.

Table 1: Performance of FBM-L, FBM-NL, and FBM-NP in Multivariate Long-term Time Series Forecasting Compared to Eight Baseline Methods on Eight Datasets.

| Method | | FBM-L | | FBM-NL | | FBM-NP | | NLinear | | PatchTST | | iTransformer | | TimeMixer | | N-BEATS | | CrossGNN | | FITS | | FreTS | |
|---|---|---|---|---|---|---|---|---|---|---|---|---|---|---|---|---|---|---|---|---|---|---|---|
| Error | | MSE | MAE | MSE | MAE | MSE | MAE | MSE | MAE | MSE | MAE | MSE | MAE | MSE | MAE | MSE | MAE | MSE | MAE | MSE | MAE | MSE | MAE |
| ETTh1 | 96 | 0.366 | 0.390 | 0.368 | 0.395 | 0.367 | 0.395 | 0.391 | 0.416 | 0.374 | 0.399 | 0.399 | 0.417 | 0.385 | 0.408 | 0.387 | 0.410 | 0.376 | 0.400 | 0.368 | 0.392 | 0.404 | 0.423 |
| | 192 | 0.403 | 0.411 | 0.408 | 0.418 | 0.407 | 0.416 | 0.421 | 0.426 | 0.417 | 0.422 | 0.436 | 0.440 | 0.429 | 0.432 | 0.428 | 0.434 | 0.419 | 0.427 | 0.404 | 0.412 | 0.461 | 0.460 |
| | 336 | 0.418 | 0.420 | 0.425 | 0.430 | 0.433 | 0.438 | 0.435 | 0.435 | 0.431 | 0.436 | 0.446 | 0.451 | 0.456 | 0.450 | 0.448 | 0.447 | 0.439 | 0.442 | 0.419 | 0.435 | 0.488 | 0.480 |
| | 720 | 0.414 | 0.438 | 0.456 | 0.466 | 0.439 | 0.459 | 0.443 | 0.457 | 0.445 | 0.463 | 0.502 | 0.503 | 0.457 | 0.462 | 0.466 | 0.471 | 0.447 | 0.465 | 0.431 | 0.458 | 0.566 | 0.553 |
| ETTh2 | 96 | 0.271 | 0.331 | 0.287 | 0.343 | 0.280 | 0.340 | 0.283 | 0.342 | 0.276 | 0.338 | 0.303 | 0.362 | 0.276 | 0.339 | 0.303 | 0.363 | 0.283 | 0.344 | 0.276 | 0.338 | 0.327 | 0.388 |
| | 192 | 0.332 | 0.373 | 0.351 | 0.386 | 0.342 | 0.382 | 0.350 | 0.387 | 0.341 | 0.378 | 0.372 | 0.403 | 0.340 | 0.381 | 0.364 | 0.402 | 0.342 | 0.387 | 0.336 | 0.377 | 0.428 | 0.450 |
| | 336 | 0.321 | 0.376 | 0.352 | 0.394 | 0.354 | 0.401 | 0.344 | 0.395 | 0.332 | 0.385 | 0.420 | 0.424 | 0.362 | 0.404 | 0.360 | 0.407 | 0.361 | 0.408 | 0.324 | 0.379 | 0.499 | 0.497 |
| | 720 | 0.369 | 0.412 | 0.397 | 0.432 | 0.386 | 0.424 | 0.395 | 0.436 | 0.379 | 0.420 | 0.420 | 0.446 | 0.398 | 0.433 | 0.428 | 0.465 | 0.423 | 0.460 | 0.373 | 0.416 | 0.727 | 0.637 |
| ETTm1 | 96 | 0.301 | 0.343 | 0.286 | 0.339 | 0.293 | 0.346 | 0.307 | 0.349 | 0.295 | 0.344 | 0.309 | 0.361 | 0.303 | 0.350 | 0.324 | 0.367 | 0.300 | 0.343 | 0.305 | 0.347 | 0.326 | 0.373 |
| | 192 | 0.337 | 0.364 | 0.324 | 0.365 | 0.334 | 0.368 | 0.347 | 0.374 | 0.333 | 0.370 | 0.345 | 0.383 | 0.356 | 0.385 | 0.363 | 0.388 | 0.335 | 0.369 | 0.338 | 0.366 | 0.359 | 0.392 |
| | 336 | 0.371 | 0.384 | 0.359 | 0.385 | 0.371 | 0.389 | 0.377 | 0.390 | 0.363 | 0.394 | 0.380 | 0.401 | 0.366 | 0.392 | 0.400 | 0.408 | 0.375 | 0.390 | 0.372 | 0.386 | 0.389 | 0.408 |
| | 720 | 0.425 | 0.415 | 0.422 | 0.424 | 0.426 | 0.420 | 0.436 | 0.425 | 0.421 | 0.420 | 0.448 | 0.442 | 0.435 | 0.434 | 0.468 | 0.448 | 0.429 | 0.420 | 0.427 | 0.416 | 0.445 | 0.441 |
| ETTm2 | 96 | 0.164 | 0.252 | 0.165 | 0.254 | 0.167 | 0.258 | 0.169 | 0.259 | 0.173 | 0.261 | 0.180 | 0.272 | 0.174 | 0.258 | 0.168 | 0.259 | 0.164 | 0.252 | 0.167 | 0.256 | 0.202 | 0.288 |
| | 192 | 0.219 | 0.290 | 0.225 | 0.296 | 0.224 | 0.296 | 0.223 | 0.294 | 0.255 | 0.306 | 0.239 | 0.311 | 0.238 | 0.300 | 0.225 | 0.301 | 0.220 | 0.294 | 0.222 | 0.293 | 0.250 | 0.322 |
| | 336 | 0.271 | 0.325 | 0.276 | 0.331 | 0.277 | 0.331 | 0.277 | 0.331 | 0.285 | 0.336 | 0.389 | 0.341 | 0.272 | 0.327 | 0.282 | 0.336 | 0.276 | 0.330 | 0.277 | 0.329 | 0.328 | 0.368 |
| | 720 | 0.364 | 0.381 | 0.365 | 0.386 | 0.367 | 0.386 | 0.371 | 0.387 | 0.365 | 0.386 | 0.374 | 0.392 | 0.368 | 0.389 | 0.376 | 0.394 | 0.372 | 0.390 | 0.366 | 0.382 | 0.431 | 0.436 |
| Electricity | 96 | 0.142 | 0.237 | 0.132 | 0.227 | 0.133 | 0.227 | 0.143 | 0.239 | 0.133 | 0.227 | 0.137 | 0.232 | 0.134 | 0.230 | 0.144 | 0.240 | 0.147 | 0.246 | 0.145 | 0.242 | 0.145 | 0.245 |
| | 192 | 0.155 | 0.248 | 0.149 | 0.243 | 0.149 | 0.242 | 0.157 | 0.250 | 0.151 | 0.244 | 0.156 | 0.249 | 0.153 | 0.245 | 0.158 | 0.252 | 0.161 | 0.258 | 0.158 | 0.253 | 0.158 | 0.255 |
| | 336 | 0.172 | 0.265 | 0.167 | 0.261 | 0.167 | 0.261 | 0.174 | 0.267 | 0.167 | 0.261 | 0.171 | 0.266 | 0.172 | 0.267 | 0.175 | 0.269 | 0.178 | 0.274 | 0.174 | 0.269 | 0.178 | 0.275 |
| | 720 | 0.212 | 0.297 | 0.207 | 0.295 | 0.208 | 0.295 | 0.214 | 0.299 | 0.210 | 0.297 | 0.195 | 0.288 | 0.212 | 0.298 | 0.217 | 0.304 | 0.214 | 0.299 | 0.213 | 0.301 | 0.220 | 0.315 |
| Traffic | 96 | 0.421 | 0.281 | 0.384 | 0.264 | 0.373 | 0.253 | 0.425 | 0.288 | 0.381 | 0.257 | 0.376 | 0.263 | 0.381 | 0.261 | 0.429 | 0.295 | 0.428 | 0.291 | 0.421 | 0.282 | 0.434 | 0.313 |
| | 192 | 0.434 | 0.286 | 0.399 | 0.269 | 0.396 | 0.266 | 0.438 | 0.291 | 0.402 | 0.270 | 0.396 | 0.274 | 0.408 | 0.273 | 0.441 | 0.299 | 0.441 | 0.295 | 0.435 | 0.288 | 0.471 | 0.311 |
| | 336 | 0.447 | 0.292 | 0.419 | 0.282 | 0.411 | 0.276 | 0.452 | 0.300 | 0.422 | 0.283 | 0.407 | 0.283 | 0.434 | 0.297 | 0.455 | 0.307 | 0.455 | 0.302 | 0.448 | 0.293 | 0.493 | 0.321 |
| | 720 | 0.477 | 0.309 | 0.448 | 0.297 | 0.442 | 0.291 | 0.482 | 0.317 | 0.454 | 0.296 | 0.449 | 0.305 | 0.469 | 0.319 | 0.486 | 0.326 | 0.486 | 0.318 | 0.478 | 0.310 | 0.535 | 0.339 |
| Weather | 96 | 0.159 | 0.207 | 0.152 | 0.199 | 0.156 | 0.204 | 0.176 | 0.226 | 0.156 | 0.206 | 0.162 | 0.211 | 0.158 | 0.204 | 0.186 | 0.238 | 0.163 | 0.227 | 0.149 | 0.198 | 0.159 | 0.218 |
| | 192 | 0.203 | 0.247 | 0.194 | 0.242 | 0.198 | 0.245 | 0.220 | 0.262 | 0.200 | 0.246 | 0.204 | 0.249 | 0.197 | 0.246 | 0.227 | 0.275 | 0.205 | 0.261 | 0.196 | 0.244 | 0.207 | 0.270 |
| | 336 | 0.252 | 0.285 | 0.244 | 0.282 | 0.248 | 0.285 | 0.265 | 0.296 | 0.252 | 0.285 | 0.252 | 0.285 | 0.242 | 0.281 | 0.274 | 0.307 | 0.250 | 0.295 | 0.245 | 0.283 | 0.252 | 0.299 |
| | 720 | 0.319 | 0.335 | 0.317 | 0.334 | 0.319 | 0.337 | 0.332 | 0.345 | 0.321 | 0.336 | 0.322 | 0.335 | 0.319 | 0.335 | 0.342 | 0.361 | 0.320 | 0.347 | 0.321 | 0.338 | 0.319 | 0.342 |
| Exchange | 96 | 0.093 | 0.211 | 0.104 | 0.226 | 0.096 | 0.196 | 0.098 | 0.219 | 0.104 | 0.227 | 0.128 | 0.254 | 0.119 | 0.247 | 0.147 | 0.274 | 0.093 | 0.211 | 0.109 | 0.235 | 0.209 | 0.350 |
| | 192 | 0.195 | 0.309 | 0.210 | 0.326 | 0.196 | 0.312 | 0.203 | 0.316 | 0.210 | 0.325 | 0.241 | 0.353 | 0.238 | 0.354 | 0.312 | 0.406 | 0.188 | 0.305 | 0.229 | 0.350 | 0.346 | 0.437 |
| | 336 | 0.347 | 0.421 | 0.398 | 0.460 | 0.353 | 0.425 | 0.356 | 0.426 | 0.366 | 0.435 | 0.393 | 0.459 | 0.417 | 0.472 | 0.522 | 0.532 | 0.363 | 0.430 | 0.400 | 0.463 | 0.634 | 0.583 |
| | 720 | 0.965 | 0.732 | 1.040 | 0.762 | 0.970 | 0.734 | 0.965 | 0.733 | 1.026 | 0.757 | 1.00 | 0.763 | 1.074 | 0.790 | 1.412 | 0.907 | 0.931 | 0.722 | 1.095 | 0.781 | 2.418 | 1.233 |
| Average | | 0.323 | 0.339 | 0.325 | 0.344 | 0.321 | 0.340 | 0.333 | 0.349 | 0.326 | 0.344 | 0.341 | 0.353 | 0.335 | 0.352 | 0.366 | 0.371 | 0.330 | 0.350 | 0.331 | 0.347 | 0.431 | 0.406 |

**FBM-L vs linear network (e.g. NLinear)**: We compare FBM-L with the NLinear model, revealing improvement across all datasets and prediction horizons, reducing the average MSE and MAE from 0.333 and 0.349 to 0.323 and 0.339, respectively. This demonstrates the effectiveness of decomposing the original time series by Fourier basis expansion. The frequency domain features allow the model to distinguish noises from hierarchical effects. We find that the improvement is more notable on datasets ETTh1 and ETTh2 due to their richer frequency components and higher prevalence of noise signals. Furthermore, our experiments reveal that even a simple vanilla linear network FBM-L has the potential to outperform state-of-the-art DNN-based architectures. More explanation is provided in Section 5.3.

**FBM-NL vs MLP-based network and FBM-NP VS Transformer-based network**: FBM-NL performs better than TimeMixer and FBM-NP performs better than PatchTST at most of the time, where the former two are both MLP-based networks and the latter two are both Transformer-based networks with patching. The experimental results further demonstrate the effectiveness of Fourier basis expansion, suggesting that extracting time-frequency features efficiently is more advantageous than relying solely on deeper architectures. It is worth noting that TimeMixer also decomposes the original time series into trend and seasonal effects, but its performance largely depends on the choice of the moving average kernel size. In contrast, our Fourier basis expansion automatically decomposes various effects hierarchically within distinct frequency levels. Although FBM-NP uses fewer patches than PatchTST, it still achieves better performance. Our FBM variants also consistently outperform other Transformer-based and MLP-based architectures, including iTransformer and N-BEATS.

**FBM variants vs. Fourier-based network**: We also observe that FBM variants make a significant improvement over all Fourier-based methods, including N-BEATS, FreTS, CrossGNN, and FITS. This discrepancy may largely be attributed to the inconsistent starting cycles and series length issues discussed in Section 3. Notably, FITS and CrossGNN also emphasize the importance of the amplitude of real and imaginary values, similar to FBM. However, they overlook the fact that Fourier basis functions are time-dependent when the input length is not divisible by a certain frequency level. Consequently, mapping in the frequency space cannot capture the time-frequency relationship effectively. For instance, CrossGNN retains only the values from the first cycle but fails to account for variations in the subsequent cycles as well as shifts in the phase of the cycles. In contrast, FBM

Table 2: Short Term Time Series Forecasting on Dataset PEMS with Forecast Horizon $L = 12$ .

| Method | FBM-L | | FBM-NL | | FBM-NP | | NLinear | | PatchTST | | TimeMixer | | iTransformer | | FiLM | | TimesNet | |
|---|---|---|---|---|---|---|---|---|---|---|---|---|---|---|---|---|---|---|
| Error | MSE | MAE | MSE | MAE | MSE | MAE | MSE | MAE | MSE | MAE | MSE | MAE | MSE | MAE | MAPE | MAE | MSE | MAE |
| PEMS04 | 0.088 | 0.195 | 0.071 | 0.172 | 0.071 | 0.170 | 0.089 | 0.196 | 0.075 | 0.179 | 0.071 | 0.173 | 0.073 | 0.176 | 0.118 | 0.237 | 0.091 | 0.196 |
| PEMS08 | 0.081 | 0.189 | 0.060 | 0.159 | 0.061 | 0.159 | 0.081 | 0.190 | 0.062 | 0.165 | 0.060 | 0.160 | 0.062 | 0.165 | 0.108 | 0.226 | 0.094 | 0.204 |
| PEMS03 | 0.077 | 0.183 | 0.060 | 0.161 | 0.061 | 0.162 | 0.078 | 0.184 | 0.065 | 0.174 | 0.062 | 0.164 | 0.061 | 0.163 | 0.108 | 0.223 | 0.089 | 0.199 |
| PEMS07 | 0.073 | 0.180 | 0.053 | 0.148 | 0.054 | 0.150 | 0.073 | 0.180 | 0.057 | 0.164 | 0.053 | 0.151 | 0.053 | 0.148 | 0.101 | 0.221 | 0.079 | 0.182 |

variants demonstrate a better representation of time-frequency domain features, as evidenced by their superior performance. In the following sections, we empirically discuss the reasons for their effectiveness through visualization.

Although our model primarily focuses on addressing LTSF challenges, we also evaluate the performance of STSF on PEMS dataset in Table 2, under the same settings as previous baseline methods. Our FBM variants consistently achieve the best performance in most cases.

### 5.3 Effect of weight on time-frequency features

In Figure 4, we visualize the effect of the weights of the one-layer vanilla linear network to show how FBM-L considers the time-frequency relationship. We decompose the weights $\mathbf{W}$ into $L$ time-specific parts, denoted as $\mathbf{W} = [\mathbf{W}_0, \mathbf{W}_1, \ldots, \mathbf{W}_{L-1}]$, where $\mathbf{W}_i$ represents the impact of the time-frequency features on the $i$-th time step of $\hat{\mathbf{Y}}$. We flatten the weights into an array similar to $\mathbf{G}$ in Figure 3, where the horizontal axis represents time, and the vertical axis represents frequency. Specifically, we visualize the heat maps of $\mathbf{W}_0$ for the ETTh1, Electricity, and Traffic datasets, comparing them with the Fourier basis. Since the datasets Electricity and Traffic are more stable than ETTh1, the weights of the linear layer for the former datasets are closer to the Fourier basis than those for the latter datasets. This implies that FBM-L considers more time-frequency relationships in ETTh1 than in Electricity and Traffic, leading to more significant improvement. The further visualization of the weights $\mathbf{W}_0$, $\mathbf{W}_{10}$, $\mathbf{W}_{20}$, $\mathbf{W}_{30}$, $\mathbf{W}_{40}$, and $\mathbf{W}_{50}$ for Electricity is provided in Figure 4 in Appendix C. It is not surprising to see that the heatmap shifts over time. In conclusion, our approach allows the model to eliminate noises across a mixture of time and frequency domains and consider time-frequency relationships. In contrast, existing studies only eliminate noises in the frequency domain.

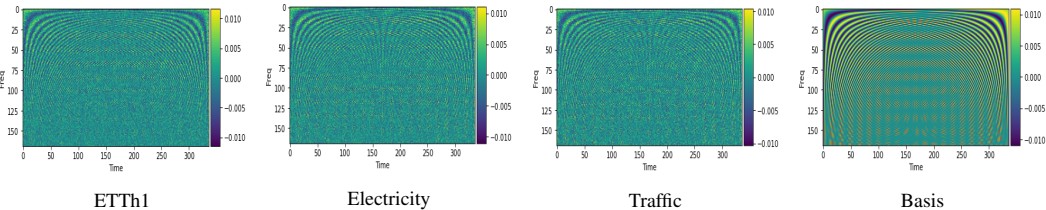

Figure 4: Effect of Weight $\mathbf{W}_0$ on Datasets ETTh1, Electricity, and Traffic.

### 5.4 Linear networks vs non-linear networks

We compare the performance of three FBM-variants on eight datasets, as shown in Table 1. FBP-NL and FBM-NP perform similarly across the datasets, as both are nonlinear. In contrast, FBM-L and FBM-NL reveal great performance differences on distinct datasets, which may be attributed to their varied data characteristics. To further explain such differences, Figure 5 shows the visualization of the frequency spectrum for a random input on the last seven dimensions across eight datasets. It shows that trending effects usually appear on high-frequency levels, seasonal effects usually fall on intermediate frequency levels, and noise signals typically manifest on low-frequency levels. This explains why the Fourier transform is beneficial for LTSF, as it hierarchically separates various effects, with specific effects (e.g., hourly, daily, weekly) aligning with their corresponding frequency levels. For instance, there is consistently high energy at the multiples of 14 on all hourly-level data ETTh1, ETTh2, Electricity, and Traffic in Figure 5. This is attributed to the day effect falling into these frequency levels, given the repeating cycle of 24 with a look-back window of 336.

By examining the frequency spectrums of all hourly-level datasets, we also find that ETTh1 and ETTh2 display a much richer frequency spectrum compared to other datasets. Figures 8 and 9 in

Appendix C further show that the frequency spectrums of Electricity and Traffic are more stable than those of ETTh1 and ETTh2. This suggests that there is a larger proportion of noise signals in datasets ETTh1 and ETTh2, but less in datasets Electricity and Traffic. The model is more likely to overfit when the data is mixed with a significant amount of noise. This is why FBM-L achieves better performance on datasets ETTh1 and ETTh2, while FBM-NL performs better on datasets Electricity and Traffic. On the other hand, the frequency spectrums for ETTm1 and ETTm2 shift to the left, contrasting with those for ETTh1 and ETTh2. This is because the granularity of ETTh1 and ETTh2 is four times larger than that of ETTm1 and ETTm2. This leads to the actual forecast horizon of the time being reduced, resulting in less noise in the data. This explains why FBM-NL performs slightly better than FBM-L on ETTm1 when the granularity changes. This also applies to datasets with other levels of granularity, such as the Exchange and Weather data. This observation validates our original hypothesis that providing the frequency spectrum is insufficient, as the real-world frequency can be ambiguous for the model when changes in granularity or series length alter the meanings of the spectrum. In conclusion, when a dataset consists of a large proportion of noise signals with a limited amount of data, a simpler network is preferred, and vice versa.

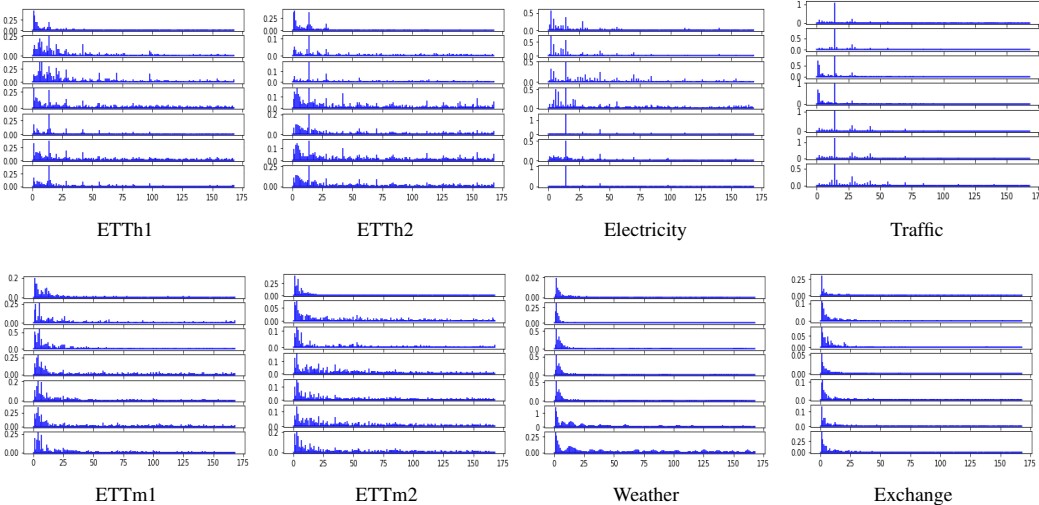

Figure 5: Frequency Spectrum of A Random Input of Last Seven Dimensions on Eight Datasets.

## 6    Conclusion

We make the first attempt to theoretically and empirically discuss several issues associated with the existing research on the deep Fourier transform for LTSF from the perspective of basis functions. Our insights and findings reveal that the real and imaginary parts of the frequency spectrum can be interpreted as the coefficients of cosine and sine basis functions, disclosing two issues commonly appearing in existing studies: inconsistent starting cycles and inconsistent series length issues. We further propose a Fourier basis mapping model Fourier basis mapping (FBM) to address these issues by extracting more efficient time-frequency features that carry explicit frequency information while retaining temporal patterns. Extensive experiments empirically verify the effectiveness of the FBM approach, to enhance various types of mapping networks and achieve state-of-the-art performance with simple one or three layers. The visualization analysis also reveals that deep neural mapping can replace the effect of inverse discrete Fourier transform and allow for more flexible consideration of time-frequency relationships. The ablation study further shows that extracting time-frequency features is more valuable than simply increasing the depth of the network. The primary focus of this study is to refine Fourier-based methods in a way that is more reasonable and interpretable. The time-frequency features hold great potential for future time series analysis tasks by enhancing various types of mapping networks (e.g., linear, MLP, Transformer networks). For example, Fourier basis expansion can enhance the performance of Transformer-based networks by only adjusting the initial projection layer after Fourier basis expansion. In future, we will also consider applying Fourier basis expansion to CNN- and RNN-based networks, as well as to tasks such as anomaly detection.

## Acknowledgments and Disclosure of Funding

This research is partly supported by NSFC (No. 62376153), Australian Research Council (ARC) Linkage Grant LP230201022, the ARC Discovery Grant DP240102050, and the ARC Council Linkage Infrastructure, Equipment and Facilities (LIEF) Grant LE240100131.

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

# A  Supplement materials for experiment results

## A.1  Datasets

We conduct our experiments on twelve real-world datasets: ETT[2] (ETTh1, ETTh2, ETTm1, ETTm2), Electricity [3], Traffic[4], Weather[5], Exchange rate (Exchange)[6] and PEMS[7] (PEMS04, PEMS08M, PEMS03, PEMS07). The granularity of ETTh1, ETTh2, Electricity, and Traffic is at an hourly time scale; the granularity of ETTm1 and ETTm2 is at a fifteen-minute time scale; the granularity of Weather is at a ten-minute time scale; the granularity of Exchange is at a daily time scale; and the granularity of PEMS is at a five-minute time scale. The ETTh datasets involve seven oil and load features of electricity transformers, spanning from July 2016 to July 2018. Traffic comprises hourly road occupancy rates measured by 862 sensors in the San Francisco Bay area from 2015 to 2016. Electricity records hourly electricity consumption (in kWh) of 321 clients from 2012 to 2014. Weather includes 21 weather indicators for 2020 in Germany, such as air temperature, humidity, and so on. Exchange tracks the daily exchange rates of eight countries from 1990 to 2016. PEMS contains the public traffic network data in California with four public subsets.

## A.2  Related work

NLinear [34] is a simple one-layer linear network with data normalization [41]. PatchTST [20] introduces a Transformer-based model for time series forecasting with two crucial improvements: patching and channel independence. FEDformer [22] adopts a structure similar to Transformer but converts time-domain features into frequency-domain features through the Fourier transform. CrossGNN [35] creates a multi-scale identifier to address noise in time series by applying the Fourier transform and utilizes graph neural network (GNN) to study cross-variable interactions. FreTS [18] projects the time series into high dimensions and characterizes the frequency space by utilizing interaction channels for the real and imaginary parts with a multilayer-perceptron-based network. TimesNet [39] transforms one-dimensional time domain information into a 2D tensor based on multiple time scales through Fourier transform, and then utilizes the convolution neural network to analyze the transformed information. N-BEATS [16] and N-HiTS [17] forecast by computing the coefficients of sine and cosine basis functions using a multilayer perceptron network. FiLM [36] combines the power of the Fourier transform with the recursive memory unit (LMU) [44], where LMU is introduced to provide a robust representation for long time series through the Legendre projection, a variant version of recurrent neural network. BasisFormer [21] computes basis functions through adaptive self-supervised contrastive learning and employs a Transformer-based architecture to study the connection between input time series and its basis functions. Autoformer [24] is a model based on Transformer that employs a decomposition architecture along with an auto-correlation mechanism to capture dependencies across time. Informer [23] introduces a probability sparse self-attention mechanism to study the temporal relationship. LogTrans [43] develops a log sparse attention mechanism to improve the efficiency of Transformer-based networks. Pyraformer [25] is a Transformer-based model that learns a multi-resolution representation of time series data using a pyramidal attention module to capture cross-time dependencies for forecasting. iTransformer [33] applies attention to study the inverted dimensions of the input sequence, capturing the dependencies between time series. TimeMixer [15] introduces a multi-scale decomposition using a bottom-up and top-down MLP network. FITS [37] proposes complex frequency linear interpolation to study the frequency domain.

## A.3  Computational efficiency

In Table 3 and 4, we evaluate the complexity of our method in comparison to other sophisticated approaches and discuss the influence of the look-back window on the final prediction results. In Table 3, we provide a comparison of the training time for one epoch cycle on the ETTh1 dataset, as well

---

[2]https://github.com/zhouhaoyi/ETDataset

[3]https://archive.ics.uci.edu/ml/datasets/ElectricityLoadDiagrams20112014

[4]http://pems.dot.ca.gov

[5]https://www.bgc-jena.mpg.de/wetter/

[6]https://github.com/laiguokun/multivariate-time-series-data

[7]https://www.kaggle.com/datasets/elmahy/pems-dataset

Table 3: Hidden Layer Complexity and Average Training Time per Epoch on the ETTh1 Dataset.

| | FBM-L | FBM-NL | FBM-NP | NLinear | PatchTST | N-BEATS | CrossGNN | iTransformer | TimeMixer | FreTS | FiLM | FITS | TimesNet |
|---|---|---|---|---|---|---|---|---|---|---|---|---|---|
| Time | 4.86 | 10.7 | 34.21 | 1.36 | 33.30 | 5.21 | 23.50 | 11.64 | 12.18 | 28.69 | 148.50 | 1.55 | 250.21 |
| Complexity | $\mathbf{O}(\frac{T^2}{2})$ | $\mathbf{O}(\frac{T^2}{2})$ | $\mathbf{O}(\frac{T^2}{P})$ | $\mathbf{O}(T)$ | $\mathbf{O}(\frac{T^2}{P})$ | $\mathbf{O}(T)$ | $\mathbf{O}(TD)$ | $\mathbf{O}(TD)$ | $\mathbf{O}(T)$ | $\mathbf{O}(T)$ | $\mathbf{O}(T^2)$ | $\mathbf{O}(T)$ | $\mathbf{O}(T^2)$ |

Table 4: The Effect of Look-back Window on ETTh1.

| Model | FBM-L | | NLinear | | PatchTST | | N-BEATS | | CrossGNN | | iTransformer | | FreTS | | FiLM | | TimeMixer | |
|---|---|---|---|---|---|---|---|---|---|---|---|---|---|---|---|---|---|---|
| Error | MSE | MAE | MSE | MAE | MSE | MAE | MSE | MAE | MSE | MAE | MSE | MAE | MSE | MAE | MSE | MAE | MSE | MAE |
| 96/96 | 0.381 | 0.394 | 0.389 | 0.401 | 0.387 | 0.401 | 0.395 | 0.406 | 0.397 | 0.410 | 0.386 | 0.405 | 0.404 | 0.417 | 0.389 | 0.398 | 0.371 | 0.399 |
| 336/96 | 0.366 | 0.390 | 0.391 | 0.416 | 0.374 | 0.399 | 0.387 | 0.410 | 0.376 | 0.400 | 0.399 | 0.416 | 0.404 | 0.423 | 0.373 | 0.397 | 0.385 | 0.408 |
| 96/192 | 0.434 | 0.421 | 0.450 | 0.437 | 0.441 | 0.431 | 0.447 | 0.433 | 0.437 | 0.429 | 0.446 | 0.438 | 0.469 | 0.455 | 0.440 | 0.425 | 0.434 | 0.429 |
| 336/192 | 0.403 | 0.411 | 0.432 | 0.421 | 0.426 | 0.422 | 0.428 | 0.434 | 0.419 | 0.427 | 0.436 | 0.442 | 0.461 | 0.460 | 0.405 | 0.416 | 0.429 | 0.432 |
| 96/336 | 0.471 | 0.438 | 0.496 | 0.462 | 0.487 | 0.455 | 0.482 | 0.447 | 0.482 | 0.449 | 0.494 | 0.463 | 0.521 | 0.486 | 0.477 | 0.441 | 0.496 | 0.451 |
| 336/336 | 0.418 | 0.420 | 0.435 | 0.435 | 0.431 | 0.436 | 0.448 | 0.447 | 0.439 | 0.442 | 0.446 | 0.451 | 0.488 | 0.480 | 0.439 | 0.422 | 0.456 | 0.450 |
| 96/720 | 0.454 | 0.453 | 0.490 | 0.478 | 0.479 | 0.471 | 0.477 | 0.467 | 0.490 | 0.478 | 0.517 | 0.500 | 0.570 | 0.546 | 0.485 | 0.473 | 0.505 | 0.486 |
| 336/720 | 0.414 | 0.438 | 0.443 | 0.457 | 0.445 | 0.463 | 0.466 | 0.471 | 0.447 | 0.465 | 0.502 | 0.503 | 0.566 | 0.553 | 0.464 | 0.473 | 0.457 | 0.462 |

as an increase in complexity in the layer. Our FBM model achieves the best prediction results at a relatively fast training speed. In contrast, attention mechanisms typically suffer from higher memory costs and slower calculation speeds. All the experiments are conducted on a GeForce GTX 3090 GPU and an Intel(R) Core(TM) i7-6850K CPU. In the past literature, a substantial amount of work selects a look-back window of 96. In Table 4, we conduct a comparison of prediction results with look-back windows of 96 and 336 on the ETTh1 dataset. Remarkably, the look-back window of 336 consistently yields better performance for almost every baseline method. While certain methods, such as iTransformer, and FreTS, exhibit lower sensitivity to the look-back window, the majority experiences a significant increase in both MSE and MAE when a look-back window of 96 is used rather than that of 336.

## B Supplemental materials for rethinking Fourier transform w.r.t. basis functions

### B.1 Proof of real-valued discrete Fourier transform via basis functions

The discrete Fourier transform (DFT) and inverse discrete Fourier transform (IDFT) can be expressed by $\frac{T}{2} + 1$ cosine basis functions and $\frac{T}{2} - 1$ sine basis functions. We provide the proof that the real-valued discrete Fourier transform can be represented by sine and cosine functions as follows:

$$
\begin{aligned}
\mathbf{H}(k) &= \sum_{n=0}^{T-1} \mathbf{X}[n] \exp\left(-i\frac{2\pi k n}{T}\right), \\
&= \mathbf{X}[0] + (-1)^k \mathbf{X}\left[\frac{T}{2}\right] + \sum_{n=1}^{\frac{T}{2}-1} \mathbf{X}[n] \exp\left(-i\frac{2\pi k n}{T}\right) + \mathbf{X}[T-n] \exp\left(i\frac{2\pi k n}{T}\right), \\
&= \mathbf{X}[0] + (-1)^k \mathbf{X}\left[\frac{T}{2}\right] + \sum_{n=1}^{\frac{T}{2}-1} \mathbf{X}[n](\cos(\frac{2\pi k n}{T}) - i\sin(\frac{2\pi k n}{T})) \\
&\quad + \sum_{n=1}^{\frac{T}{2}-1} \mathbf{X}[T-n](\cos(\frac{2\pi k n}{T}) + i\sin(\frac{2\pi k n}{T})), \\
&= \mathbf{X}[0] + \cos(\pi k)\mathbf{X}\left[\frac{T}{2}\right] + \sum_{n=1}^{\frac{T}{2}-1} (\mathbf{X}[n] + \mathbf{X}[T-n])\cos(\frac{2\pi k n}{T}) \\
&\quad - i\sum_{n=1}^{\frac{T}{2}-1} (\mathbf{X}[n] - \mathbf{X}[T-n])\sin(\frac{2\pi k n}{T}), \quad k = 0, \ldots, T-1.
\end{aligned}
\tag{6}
$$

Then, we can deduce that the frequency spectrum is Hermitian symmetric, where $H(k) = \overline{H(T-k)}$.

## B.2 Proof of real-valued inverse discrete Fourier transform via basis functions

The real-valued inverse discrete Fourier transform can be expressed by sine and cosine basis functions as follows:

$$
\begin{aligned}
\mathbf{X}[n] &= \frac{1}{T} \sum_{k=0}^{T-1} \mathbf{H}[k] \exp\left(i \frac{2\pi kn}{T}\right) = \sum_{k=0}^{\frac{T}{2}} \mathbf{H}[k](\cos(\frac{2\pi kn}{T}) + i \sin(\frac{2\pi kn}{T})) \\
&+ \sum_{k=1}^{\frac{T}{2}-1} \mathbf{H}[T-k](\cos(\frac{-2\pi kn}{T}) + i \sin(\frac{-2\pi kn}{T}))), \\
&= \frac{1}{T}(\mathbf{H_R}[0] + \mathbf{H_R}[\frac{T}{2}] \cos(\pi n) + \sum_{k=1}^{\frac{T}{2}-1} (\mathbf{H_R}[k] + i\mathbf{H_I}[k])(\cos(\frac{2\pi kn}{T}) + i \sin(\frac{2\pi kn}{T})) \\
&+ \sum_{k=1}^{\frac{T}{2}-1} (\mathbf{H_R}[k] - i\mathbf{H_I}[k])(\cos(\frac{2\pi kn}{T}) - i \sin(\frac{2\pi kn}{T}))), \\
&= \frac{2}{T} \sum_{k=1}^{\frac{T}{2}-1} \left( \mathbf{H_R}[k] \cos\left(\frac{2\pi kn}{T}\right) - \mathbf{H_I}[k] \sin\left(\frac{2\pi kn}{T}\right) \right) + \frac{1}{T}(\mathbf{H_R}[0] \\
&+ \mathbf{H_R}[\frac{T}{2}] \cos(\pi T)), \\
&= \frac{1}{T} \sum_{k=0}^{\frac{T}{2}} \left( a_k \cos\left(\frac{2\pi kn}{T}\right) - b_k \sin\left(\frac{2\pi kn}{T}\right) \right), \quad n = 0, \ldots, T-1,
\end{aligned}
$$

$$
a_k = \begin{cases} \mathbf{H_R}[k], & k = 0, \frac{T}{2} \\ 2 \cdot \mathbf{H_R}[k], & k = 1, \ldots, \frac{T}{2} - 1 \end{cases} \quad b_k = \begin{cases} \mathbf{H_I}[k] = 0, & k = 0, \frac{T}{2} \\ 2 \cdot \mathbf{H_I}[k], & k = 1, \ldots, \frac{T}{2} - 1. \end{cases}
\tag{7}
$$

## B.3 Equivalence between the IDFT and N-BEATS-Seasonal-Block design for frequency side mapping.

We provide further explanation of why IDFT and N-BEATS-Seasonal-Block are equivalent for the frequency space mapping. The general structure of a Fourier-based method can be summarized as follows:

$$
\begin{aligned}
\mathbf{H}^X &= DFT(\mathbf{X}), \\
\mathbf{H}^Y &= Encoder(\mathbf{H}^X), \\
\hat{\mathbf{Y}} &= IDFT(\mathbf{H}^Y).
\end{aligned}
\tag{8}
$$

The encoder can refer to any deep neural architecture, such as a CNN-based network, RNN-based network, GNN-based network, Transformer-based network, or MLP-based network. Therefore, the model aims to make forecasting by mapping $\mathbf{H}^X$ to $\mathbf{H}^Y$. However, if we rewrite the IDFT in Equation (8) by Equation (4) and assume $\mathbf{M} = [0, \ldots, L-1]$, we can obtain the following equation:

$$
\hat{\mathbf{Y}} = \frac{1}{L} \sum_{k=0}^{\frac{L}{2}} \left( a_k \mathbf{H_R}[k]^Y \cos\left(\frac{2\pi k\mathbf{M}}{L}\right) - a_k \mathbf{H_I}[k]^Y \sin\left(\frac{2\pi k\mathbf{M}}{L}\right) \right),
$$

$$
a_k = \begin{cases} 1, & k = 0, \frac{L}{2} \\ 2, & k = 1, \ldots, \frac{L}{2} - 1. \end{cases}
\tag{9}
$$

This is equivalent to the N-BEATS-Seasonal-Block design, as their structure can be expressed as follows:

$$
\boldsymbol{\theta} = MLP(\mathbf{X}),
$$

$$
\hat{\mathbf{Y}} = \sum_{i=0}^{\frac{L}{2}} \cos\left(\frac{2\pi i\mathbf{M}}{L}\right) \boldsymbol{\theta}_i + \sin\left(\frac{2\pi i\mathbf{M}}{L}\right) \boldsymbol{\theta}_{i+\frac{L}{2}+1}.
\tag{10}
$$

Table 5: The hyperparameter of FBM-NL, FBM-NP and PatchTST

|  |  | ETTh1 | ETTh2 | ETTm1 | ETTm2 | Electricity | Traffic | Weather | Exchange | PEMS |
|---|---|---|---|---|---|---|---|---|---|---|
| FBM-NL | $H_1/H_2$ | 1440/1440 | 1440/1440 | 1440/1440 | 1440/1440 | 1440/1440 | 1440/1440 | 1440/1440 | 1440/1440 | 720/360 |
| FBM-NP | $P/K$ | 14/16 | 14/16 | 14/128 | 14/128 | 14/128 | 28/128 | 14/128 | 14/128 | 14/128 |
| PatchTST | $P/K$ | 42/16 | 42/16 | 42/128 | 42/128 | 42/128 | 42/128 | 42/128 | 42/128 | 42/128 |

In the N-BEATS model, they use a multilayer perceptron to compute the coefficients of sine and cosine waves, denoted as $\boldsymbol{\theta}$. Although the dimension of $\boldsymbol{\theta} \in \mathbf{R}^{L+2}$ is twice as large as the dimension of $\mathbf{H}^Y \in \mathbf{C}^{L/2+1}$, if we consider the real part and imaginary part of $\mathbf{H}^Y$ separately, then the dimension is exactly the same. As a result, the objective of the deep Fourier-based method and the N-BEATS model in Equation (9) and Equation (10) are equivalent: to compute the coefficients of sine and cosine basis functions with tiered frequency, as the scalar of a basis function doesn't provide any difference in deep learning. The only difference is that the former one maps the frequency domain to the frequency domain, and the latter one maps the time domain to the frequency domain.

## C  Supplemental results to support experimental analysis

Figure 6 is provided as supplementary material for Section 4. In Figure 6, FBM-L represents a linear network, FBM-NL is a three-layer MLP network with dropout and ReLU as the activation function, and FBM-NP has the same structure as PatchTST but differs in the number of patches and initial projection layer. All the rest hyperparameters for FBM-NP are the same as those for PatchTST. The key hyperparameters are provided in Table 5. You can always tune them to achieve better results than us. Since validation set is used to determine when to early stop the training, adjusting hyperparameters based on the test set can undermine generalization. Additionally, tuning random seeds to achieve better results may compromise reproducibility across different devices. Therefore, we use fixed hyperparameters in most cases and maintain the same random seed throughout the entire experiment.

Figure 7 is provided as supplementary material for Section 5.3. In Figure 7, we visualize the weights $\mathbf{W}_0$, $\mathbf{W}_{10}$, $\mathbf{W}_{20}$, $\mathbf{W}_{30}$, $\mathbf{W}_{40}$, and $\mathbf{W}_{50}$ for the Electricity dataset and observe that the heatmaps of weights shift over time.

Figures 8 and 9 are provided as supplementary material for Section 5.4. In these figures, we provide the frequency spectrums of three random inputs for each dataset. We observe that the frequency spectrums of the Electricity and Traffic datasets are quite stable, followed by the ETTm1, ETTm2, and Weather datasets. In contrast, the frequency spectrums of ETTh1 and ETTh2 are not stable, as reflected by the changing landscape over time with less similarity at certain levels.

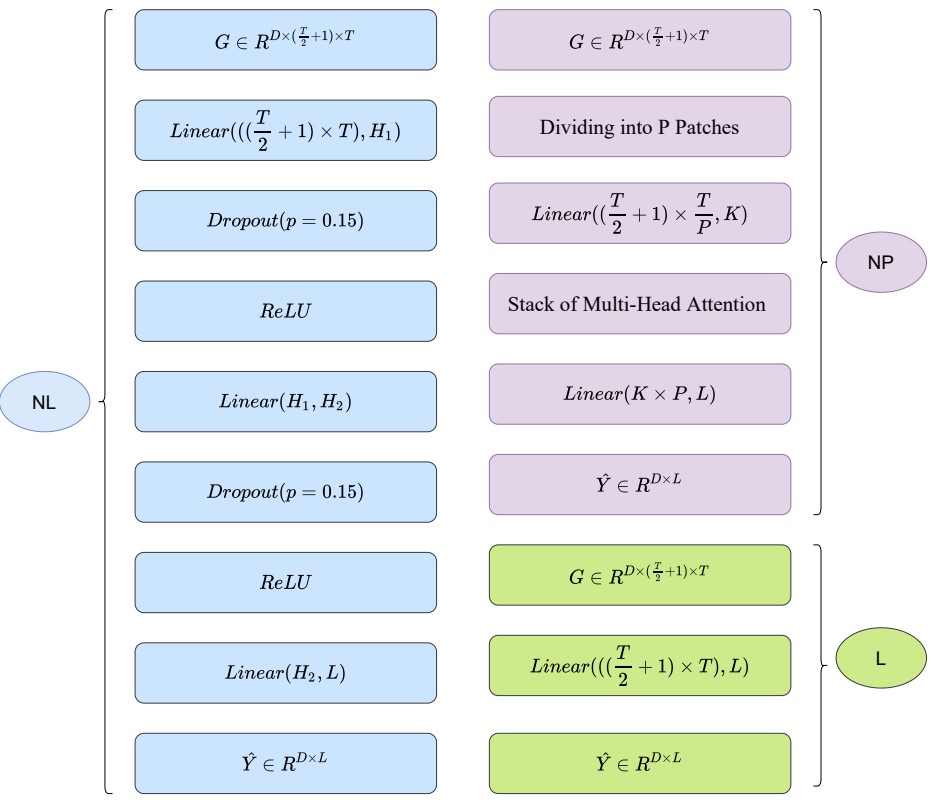

Figure 6: The Layers of FBM-L, FBM-NL, and FBM-NP in Detail. FBM-L represents a vanilla linear network, FBM-NL denotes a three-layer non-linear network, while FBM-NP shares the same structure as PatchTST but it differs in the number of patches and applies the patching after the Fourier basis expansion.

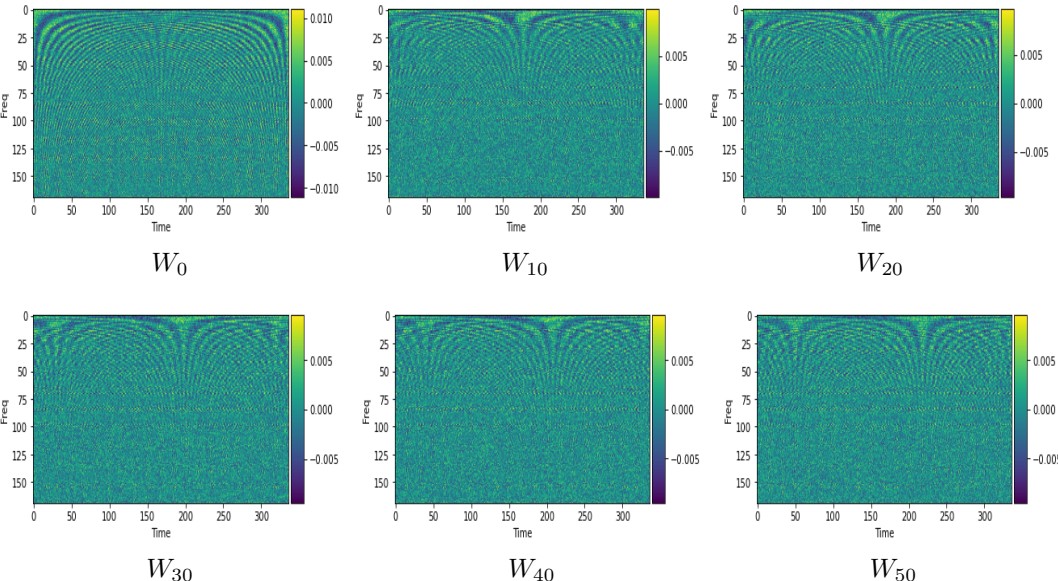

Figure 7: The Visualization of Weights $W_0$, $W_{10}$, $W_{20}$, $W_{30}$, $W_{40}$, and $W_{50}$ on the Electricity Dataset.

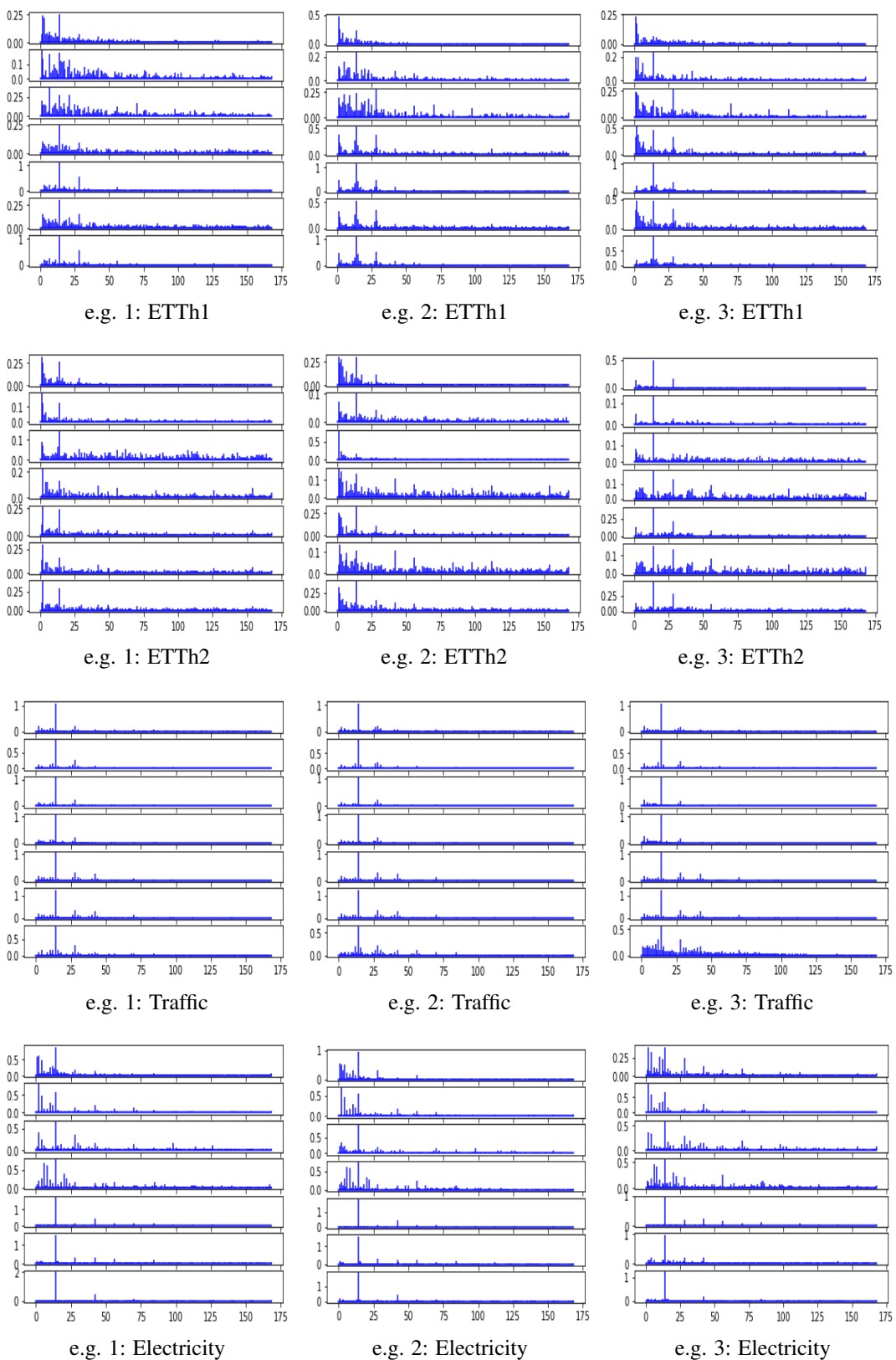

Figure 8: The Frequency Spectrum of Three Extra Random Inputs of the Last Seven Dimensions on the ETTh1, ETTh2, Traffic and Weather Datasets.

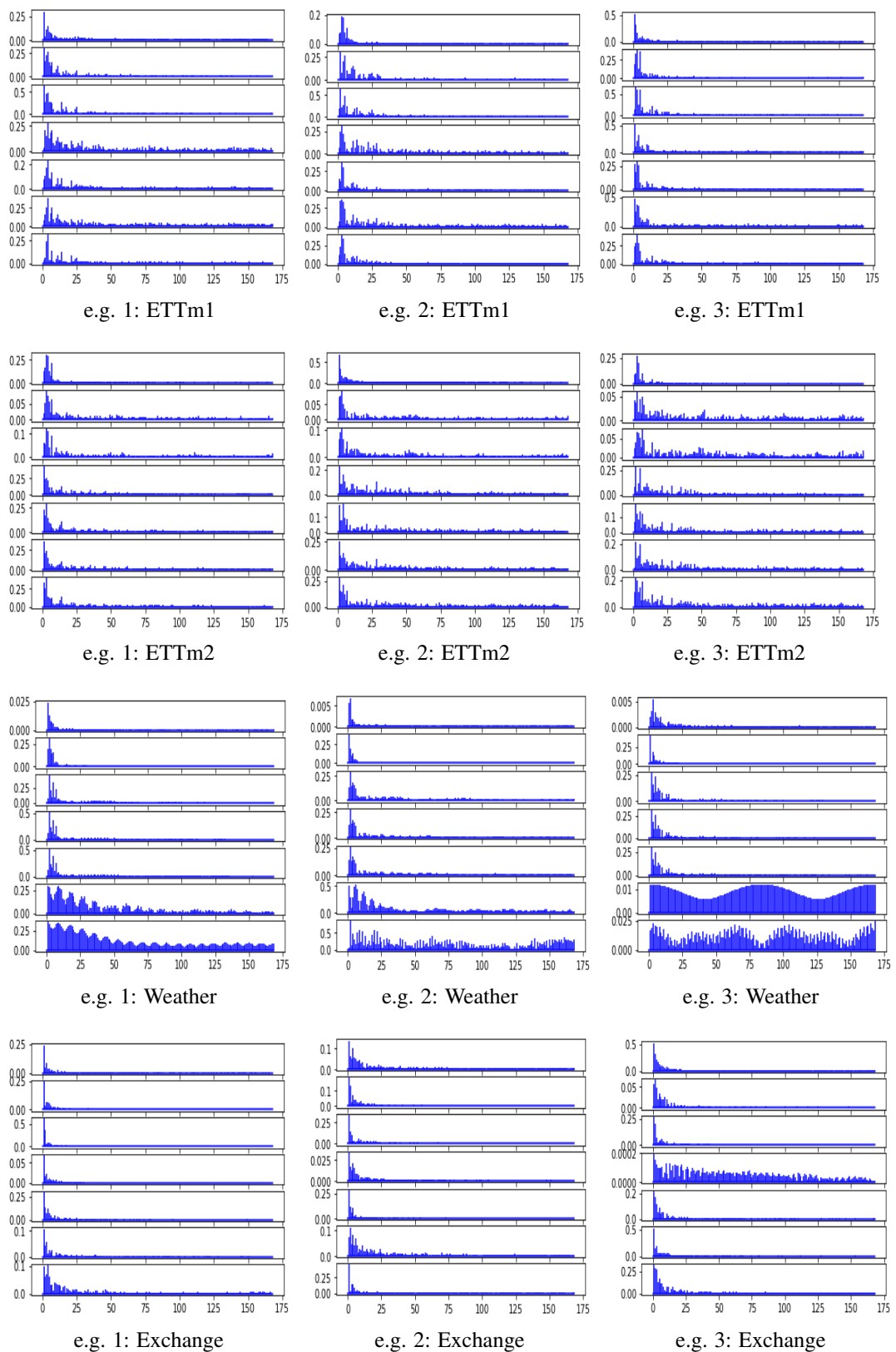

Figure 9: The Frequency Spectrum of Three Extra Random Inputs of the Last Seven Dimensions on the ETTm1, ETTm2, Electricity and Exchange Datasets.

