# OpenReview forum: "Rethinking Fourier Transform from A Basis Functions Perspective for Long-term Time Series Forecasting"
_NeurIPS.cc/2024/Conference — NeurIPS 2024 poster_

### Official Review · Reviewer_8ZRG · 2024-06-23

**Soundness:** 3
**Presentation:** 2
**Contribution:** 2
**Rating:** 2
**Confidence:** 4

**Summary:**

The authors propose a Fourier basis mapping model, named FBM, for long-term time series forecasting. FBM embeds the discrete Fourier transform with basis functions, and then introduces a mapping network to replace the inverse discrete Fourier transform. This approach allows FBM to capture implicit frequency features while preserving temporal characteristics. Experiments demonstrate the effectiveness of the proposed framework.

**Strengths:**

(1)The paper is easy to follow.

(2)The experimental results on eight real-world datasets prove that FBM achieves competitive performance.

**Weaknesses:**

(1)The technical contributions are incremental and the overall impact is not significant enough, as some key components are similar to existing works. For example, the Fourier basis mapping is similar to FEDformer. In addition, the three mapping methods used in the decoder look like a simple combination of linear network, nonlinear MLP, and PatchTST.

(2)The introduction of the model architecture is confusing, as many implementation details of the modules in Figure 1 and Figure 3 are not clearly explained or even introduced (e.g., the Fourier basis expansion operation, the normalization operation, and the de-normalization operation).

(3)Several groups of comparative experiments are carried out, but some SOTA works [1, 2, 3] are not compared or mentioned. The authors should compare their model with SOTA works to demonstrate its effectiveness.

(4)There are many typos and writing mistakes in the manuscript. For example, on page 2, "Huange et. al." should be "CrossGNN". On page 6, "Table 2" should be "Table 1". On page 12, "Crossgnn" should be "CrossGNN", "N-beats" should be "N-BEATS", and "Nhits" should be "N-HITS". The manuscript requires a thorough proofreading.

(5)Some figures provided are too small to allow for a thorough investigation of the details. The authors should use vector figures instead. In addition, in Figure 5, the meaning of the x- and y-axes need to be explained and the meaning of the number "5" in the right part should also be clarified.

[1] Liu Y, Hu T, Zhang H, et al. iTransformer: Inverted Transformers Are Effective for Time Series Forecasting. ICLR 2024.
[2] Wang S, Wu H, Shi X, et al. Timemixer: Decomposable multiscale mixing for time series forecasting. ICLR 2024.
[3] Chen P, Zhang Y, Cheng Y, et al. Pathformer: Multi-scale transformers with Adaptive Pathways for Time Series Forecasting. ICLR 2024.

**Questions:**

See Weaknesses.

**Limitations:**

No limitations or potential negative societal impacts are discussed in the paper.

---

> ### Author Rebuttal · Authors · 2024-08-07
>
> We appreciate your constructive suggestions, which will be fully reflected in the final version, including clarifying misreading/misunderstanding. Extra experiment results and the updated Figure 1 are attached in the one-page PDF within the global response for all reviewers.
>
> **Weakness (1)**
>
> There could be misreading/misunderstanding of the motivation and main contributions of our work. We redraw Figure 1 and revise Introduction as explained in **Introduction and Figure 1** of the global response to clarify the workflow and compare our FBM with others. Our contribution is to extract more interpretable features to enable much more effective downstream mapping.
>
> Thus, we disagree with the statement that “some key components are similar to existing works.” Instead, our FBM opens a new perspective for time series forecasting using the FBM with the time-frequency features, making the following main differences:
>
> 1.	The Existing Fourier-based mapping is in frequency space, while FBM is conducted in time-frequency space.
> 2.	Existing methods face inconsistent starting cycles and inconsistent series length issues, addressed by FBM.
> 3.	Existing methods ignore that Fourier basis could be time dependent, addressed by FBM.
>
> In fact, existing Fourier-based mapping methods (e.g. FEDformer) do not involve Fourier basis mapping like ours. They use real and imaginary parts as inputs to their mapping networks and conduct in frequency space. In contrast, FBM conducts mapping in both time and frequency space.
>
> We provide a new perspective that real and imaginary parts can be interpreted as the coefficients of cosine and sine basis functions at different frequencies, as shown in Eq. (3). However, existing Fourier-based mappings do not involve such basis functions, thus failing to interpret these coefficients correctly. Eq. (4) shows that adding a cosine wave and a sine wave at the same frequency results in a shifted cosine wave of the same frequency. This means that the crucial information is embedded in the amplitude and phase of each cycle rather than in the real and imaginary parts. This leads to issues of inconsistent starting cycles and inconsistent series lengths in existing methods, addressed by our FBM.
>
> In addition, a Fourier basis function is time-dependent when the input length is not divisible by a frequency level (see new Figure 1 in PDF ). Consequently, the mapping in the frequency space fails to capture this time-dependent frequency information. Our FBM addresses these issues. It decomposes an input time series into $\frac{T}{2}+1$ pieces at hierarchical frequency granularity, where T is the input sequence length. We show that time-frequency features enable simplified downstream mapping, and very simple mapping networks (L-vanilla linear network and NL-three-layer MLP) can beat complicated Transformers, etc. FBM achieves SOTA performance using either a vanilla linear layer or a three-layer MLP on eight real-world datasets for LTSF and PEMS datasets for STSF.
>
> We sincerely appreciate if you could read our response to Weakness W2 for Reviewer akVH as well, as it is important to understand the unique contribution of time-frequency features.
>
>
> **Weakness (2)**
>
> Please refer to new Figure 1 and the updated Introduction in the global response.
>
> As discussed in line 167, we use the normalization and denormalization same as the following reference:
>
> Ref: Reversible Instance Normalization for Accurate Time-Series Forecasting against Distribution Shift.
>
> Specifically, normalization subtracts the average value of the input time series and denormalization adds that back to the forecast time series at the end, as widely used in LTSF and demonstrated effective by the reference.
>
> The Fourier basis expansion operation follows Eq. (3) and aims to solve the issues discussed in Section 3 of the paper.
>
> The internal mechanisms of the three mapping methods are provided in Figure 6 in Appendix.
> Assuming the look-back window $T=336$ and forecast horizon $L=96$, FBM-L decomposes the time series into $\frac{T}{2}+1$ pieces at different frequency granularities by Eq (3). Thus, our FBM-L mapping uses a “nn.Linear (336 $\times$ 169, 96)”, and the FBM-NL includes two additional nn.Linear layers with activation functions. FBM-NL uses the same structures and hyperparameters for each dataset across different horizons for better reproducibility.
>
> **Weakness (3)**
>
> Four SOTA baseline methods Pathformer, iTransformer, TimeMixer and TimesNet [1] are compared in Table 2 of the PDF for the global response. Due to time constraints, we only finish LTSF experiments on datasets ETTh1 and ETTh2 but also add extra experiments for STSF on the PEMS dataset. FBM-L and FBM-NL consistently achieve the SOTA performance for both LTSF and STSF without tuning, as shown in the PDF.
>
> **Weakness (4)**
>
> Many thanks for pointing out the typos, we’ve revised the paper and thoroughly polished the paper for the final version.
>
> **Weakness (5)**
>
> Figure 1 is updated in the PDF to emphasize the distinctive features of FBM against existing methods, pointing out the issues of previous methods and highlighting our main contribution. All other figures are adjusted for better visualization. In Figure 5, the x-axis represents the frequency level (Hz), and the y-axis represents the amplitude of real and imaginary values. ``5’’ refers to the last value of 175, removed from all figures.
>
> **Limitations**:
>
> This paper on time series forecasting does not involve negative social impact.
>
> Ref:
>
> [1] TimesNet: Temporal 2D-Variation Modeling for General Time Series Analysis
>
> We have thoroughly addressed all comments and produced a new version of the paper and thank you for any further kind suggestions.

---

> > ### Comment · Reviewer_8ZRG · 2024-08-12
> >
> > Thanks for the rebuttal. I read all of the authors' responses and the discussions with other reviewers.
> >
> > (1) Although the authors have added several comparative experiments with latest baselines, almost all the reproduced results differ significantly from those in the original paper, making the results unreliable. For example, '0.407 0.423' should be '0.386 0.405' for iTransformer on ETTh1 dataset under 96 forecasting horizon and '0.176 0.354' should be '0.078 0.183' for iTransformer on PEMS04 dataset under 12 forecasting horizon.
> >
> > (2) The authors have claimed that the main difference between FBM and existing Fourier-based mapping is that "FBM is conducted in the time-frequency space." However, I did not see any targeted design towards "time-frequency space" in Section 4. In addition, there are many key concepts (e.g., Time Dependent, L is not divisible by K, and Time duration) in **Figure 1** of $\underline{\text{global response}}$ that are not adequately described or even explained in the main paper, which makes it difficult to understand in some places.
> >
> > (3) The authors have promised to "revise the paper and thoroughly polish the paper for the final version", however, in **Figure 1** of $\underline{\text{global response}}$, we still find many inconsistency and informal expressions, which reflect a lack of rigor and carelessness of authors in the scientific research:
> >
> > * "Frequency level" should be "Frequency Level";
> >
> > * "Time duration" should be "Time Duration";
> >
> > * "FBM's" should be "Our Method".
> >
> >
> > Given the above question and the writing, I think this paper is not ready for being published in NeurIPS, and I have updated my scores.

---

> ### Comment · Reviewer_Z6Ax · 2024-08-07
>
> I think you should see the author's response to my question. "**Our main contribution is on extracting better initial features enable more effective downstream mapping. With time-frequency features, very simple mapping networks (L-vanilla linear network and NL-three-layer MLP) can achieve the SOTA in any circumstance. **The internal mechanisms of FBM variants are provided in Figure 6 in Appendix. Assuming the look-back window and forecast horizon , FBM-L decomposes the original time series into pieces at different frequency granularities. Thus, our FBM-L mapping uses a “nn.Linear (336 169, 96)”, and the FBM-NL includes two additional nn.Linear layers with activation functions, see Figure 6 in Appendix for details." **This paper is cool because it can improve the simple method's performance.**

---

> ### Author Response · Authors · 2024-08-12
> **Reponses to your points - 1**
>
> We appreciate your response and thank you for taking your valuable time to read our previous responses. Below, we address your three new points, and clarify potential misreadings. The response is a bit lengthy, please read this multiple-page response titled **Reponses to your points – 1** to **Reponses to your points – 3** due to the space limit, to substantially address your comments. We appreciate your valuable time and patience in reading this response.
>
> **Response to your point (1)**
>
> First, **misreading of different settings and evaluation measures**
>
> You misread or overlook the different input sequence lengths used by our method (**with length 336**) vs that by iTransformer (**with length 96**) on dataset ETTh1, and the different performance measures for ours (**per MAE and MAPE**) vs iTransformer (**per MSE and MAE**) on dataset PEMS04.
>
> The result of ‘0.407 0.423’ corresponds to the input sequence length of 336, while ‘0.386 0.405’ corresponds to the sequence length of 96 for iTransformer on ETTh1 under 96 forecasting horizons. This setting difference applies to all new experimental comparisons in Table 2 in the PDF file to all reviewers.
>
> The result of '0.176 0.354' refers to the value of MAE and MAPE, respectively; in contrast, '0.078 0.183' refers to the values of MSE and MAE, respectively for iTransformer on dataset PEMS04 under 12 forecasting horizons. This setting difference applies to all experiment results in Table 3 in the PDF file.
>
> In conclusion, 1) the performance difference you misread for iTransformer on ETTh1 corresponds to different input sequence lengths, i.e., we use 336 versus 96 used in the original paper; 2) the performance difference you misread for iTransformer on PEMS04 correspond to different evaluation measures, i.e., MAE and MAPE for ours vs MSE and MAE for iTransformer.
>
> Please also refer to our previous response **Q1** in the second round of response titled **Further Responses to Your New Questions Q1 and Q5** to Reviewer akVH.
>
>
> Second, **we explain why we use length 336 rather than 96 as in iTransformer.**
>
> The input sequence length of 336 always produces better results than 96. This has been verified by the following studies, where methods in Group A with sequence length 336 consistently report lower MSE and MAE values than those in Group B with sequence length 96.
>
> Group A:
>
> 1.	PatchTST: A Time Series is Worth 64 Words: Long-term Forecasting with Transformers
> 2.	NLinear: Are Transformers Effective for Time Series Forecasting
>
> Group B:
>
> 1.	Pathformer: Multi-scale Transformers with Adaptive Pathways for Time Series Forecasting
> 2.	iTransformer: Inverted Transformers Are Effective for Time Series Forecasting
> 3.	TimeMixer: Decomposable Multiscale Mixing for Time Series Forecasting
> 4.	TimesNet: Temporal 2D-Variation Modeling for General Time Series Analysis
>
> The above difference between our reported results and that in iTransformer is actually consistent with what we can identify in the results of iTransformer in comparison with PatchTST.
>
> For example, if you compare the MSE and MAE scores reported in PacthTST with the scores reported by iTransformer for PatchTST, you will find even more significant difference. We collect and present their results in the following table:
>
> PatchTST’s results | In PatchTST's paper with T=336 | In PatchTST 's paper with T=336 | In iTransformer's paper with T=96 | In iTransformer's paper with T=96
> -|-|-|-|-
> Error|MSE| MAE| MSE| MAE|
> ETTh1 L= 96|0.375|0.399|0.414|0.419
> ETTh1 L=192|0.414|0.421|0.460|0.445
> ETTh1 L=336|0.431|0.436|0.501|0.466
> ETTh1 L=720|0.449|0.466|0.500|0.488
>
>
> Therefore, we set the input sequence length to 336. Accordingly, our experiment settings are the same as that of PatchTST for better performance, which are different from Pathformer, iTransformer, TimeMixer, and TimesNet using the input sequence length of 96.
>
> Third, **we explain why we use MAE and MAPE rather than MSE and MAE as in iTransformer.**
>
> For the PEMS04 dataset, MAE and MAPE are more suitable metrics than MSE and MAE, as shown in other baselines like TimeMixer. The difference in MAPE between our results and TimeMixer's is due to their filtering of extremely large values, while we retain the original values. However, both comparisons are reasonable.
>
> In fact, our experiments also verify the above observation with a lower MAE score for the input sequence length of 336 than that reported in the iTransformer paper for length 96 (**0.176 vs 0.183**). This result further demonstrates that the input sequence length of 336 is more robust.

---

> > ### Comment · Reviewer_8ZRG · 2024-08-13
> >
> > Thanks again for your detailed rebuttal. I read all of the authors' responses and the discussions with other reviewers.
> >
> > (1) The authors claim that they use MAPE as the evaluation metric on PEMS04 dataset. However, **the baseline methods (e.g., iTransformer and TimeMixer) do not use MAPE as the evaluation metric.** Simply stating that 'MAE and MAPE are the most reliable metrics for the evaluation on the PEMS04 dataset' lacks convincing.
> >
> > (2) The authors claim that their input length is 336. I set the input length to 336 and rerun the source code of iTransformer with the default parameters on ETTh1 dataset. However, as reviewer **akVH** points out, **the results still show a significant difference from what the authors reported.**
> >
> > (3) The authors claim that their input length is 336, which is similar to PatchTST. Why the authors do not directly use the experimental results from PatchTST? In addition, I rerun the source code of PatchTST and the results are generally consistent with those reported in the paper. **However, the results still differ significantly from those in the authors' paper**. For example, on Weather dataset under 96 forecasting horizon, the PatchTST results in the original paper are $\underline{\text{0.152 0.199}}$, while in the authors' paper, the PatchTST results are $\underline{\text{0.176 0.226}}$, and the authors' own results are $\underline{\text{0.152 0.199}}$. As reviewer **akvH** points out, this may lead to ambiguity.

---

> > > ### Author Response · Authors · 2024-08-13
> > > **Response to Your New Points (1), (2), and (3)**
> > >
> > > We appreciate your response and thank you for taking your valuable time to read our responses again.
> > >
> > > **Response to your new point (1)**
> > >
> > > TimeMixer does use MAE and MAPE as the evaluation metrics for the PEMS04 dataset (see Table 3 in their paper).
> > >
> > > **Response to your new point (2)**
> > >
> > > We don't know whether Reviewer akVH ran his/her experiments with an input length of 336 or 96, as he/she hasn’t stated the experimental settings or provided the exact scores. This makes it difficult for us to fully address this question. However, we have done our best to reproduce every method, and it’s common to encounter reproduction errors, especially when experiments are conducted on different devices and settings. Even a tiny, unnoticed variation in the experimental setup can lead to significant differences in results. Additionally, since all baseline methods were run only once, some variation is to be expected. It’s also worth mentioning that even if we had achieved the results reported in the iTransformer paper, they are still not as good as ours (ETTh1-L=96: iTransformer reports MSE: 0.386, MAE: 0.405; Our FBM-L reports MSE: 0.366, MAE: 0.390).
> > >
> > > **Response to your new point (3)**
> > >
> > > In line 442 of our paper, we stated that we split the ETT dataset into 12/4/4 months and the other datasets into training, validation, and test sets with a ratio of 6.5/1.5/2. However, PatchTST splits the Weather dataset with a ratio of 7/1/2. We chose our splitting method because a validation set that is only half the size of the test set is relatively too small, particularly in comparison to the ETT dataset. Consequently, the experimental settings for the Weather dataset with a 96-step forecasting horizon differ as well. Although reproduction errors are common under different experimental settings, when running on different devices, and due to error bar in baseline methods, we have made every effort to reproduce each baseline method as accurately as possible and ensure a fair comparison. In the final version of the paper, we will run multiple iterations for each baseline method and report the average values.
> > >
> > > We hope our explanation has addressed your concerns, and we appreciate your valuable time spent reviewing our response. We welcome any further comments or suggestions to help improve our work. Thank you very much.

---

> ### Author Response · Authors · 2024-08-12
> **Reponses to your points - 2**
>
> Lastly, **you are welcome to verify our codes for the above clarification of your misreading**
>
> Regarding the above different evaluation settings and resultant performance, you are more than welcome to verify our shared codes in the **Supplementary Material** (in line 524 in the paper and our global response to all reviewers) to verify our results. Our codes follow the same structure as iTransformer, so it is easy to run if you are familiar with it. Once you have chance to test our codes, you would verify our above clarification and identify your misreading as well. You could verify FBM-L on the ETTh1 and ETTh2 datasets first, as it runs very quickly.
>
>
> **Response to your point (2)**
>
> Regarding the design for the ‘time-frequency space’, this is in fact the key innovation of our work, and we have substantially explained how to design and implement it in different ways and places.
>
> First, **Figure 3 explains the FBM structure**
>
> In Section 4, we provide Figure 3 to summarize the framework of our proposed Fourier basis mapping (FBM), and explain its working mechanism, copied below for your reading convenience (Lines 160-166):
>
> To address the two issues, we introduce the Fourier basis mapping model FBM. Figure 1 shows the architecture of FBM. The key strength of FBM lies in its designs of generating more implicit frequency features while retaining temporal features, as the actual frequency is inferred by the model through basis functions. The process is analogous to decomposing the original time series into $\frac{T}{2}+1$ components with tiered frequency levels, which allows the model to separate various effects hierarchically but identify the affiliated noises. Consequently, FBM emerges as a more natural method to extract the mixture of both time and frequency domain features.
>
> Second, this section further **elaborates the design for capturing time-frequency features**, e.g., as shown in Lines 170-172, and Lines 173-176:
>
> **Next, we multiply the real part of $\mathbf{H}$ (denoted as $\mathbf{H_R}$) with the cosine basis $\mathbf{C}$ and the imaginary part of $\mathbf{H}$ (denoted as $\mathbf{H_I}$) with the sine basis $\mathbf{S}$ to obtain the mixture of frequency and temporal features.
> …
> Further, we combine the cosine and sine basis functions to obtain the shifted starting cycle of the time series, forming $\mathbf{G}$. The scalar $2$ is derived in Eq. (4) to ensure that, if we sum along the $\frac{T}{2}+1$ frequency domain, we can obtain the original time series without corrupting the time domain information.**
>
> Third, we further **explain the mechanism in this section**, e.g.:
>
> In Line 163: “The process is analogous to decomposing the original time series into $\frac{T}{2}+1$ components with tiered frequency levels, which allows the model to separate various effects hierarchically but identify the affiliated noises.”
>
> In Line 165: “Consequently, FBM emerges as a more natural method to extract the mixture of both time and frequency domain features.”
>
> In Line 173: “Further, we combine the cosine and sine basis functions to obtain the shifted starting cycle of the time series, forming G.”
>
> In Line 177, “…it allows the model to eliminate noise along both time and frequency domains.
> Finally, we use a decoder to map the features G to the output time series.“
>
> Lastly but not least, we have **updated Figure 1 in the PDF file** to all reviewers, where the time-frequency space is shown in the lower middle panel, extracted from input and further fed to the mapping networks.
>
> In fact, we have revised the introduction per the suggestion by **Reviewer Z6Ax** to explain the main insights and innovation of our work, who has accepted our revision. For your reading convenience, we copy the two paragraphs in our next response.

---

> ### Author Response · Authors · 2024-08-12
> **Reponses to your points - 3**
>
> **Here is the revised last two paragraphs of Introduction:**
>
> Lastly, Fourier-based time series modeling emerges as a new paradigm to remove noise signals by considering diverse effects hierarchically at different frequency levels. However, methods like FEDformer, FreTS, FiLM, FGNet, and FL-Net use real and imaginary parts as inputs to their mapping networks but cannot easily interpret their coefficients because the crucial information is stored in the amplitude and phase of each cycle. This leads to inconsistent starting periods and series lengths, which are often ignored in existing research. Other methods e.g., CrossGNN and TimesNet use the top-k amplitudes to filter noises. However, a higher amplitude does not necessarily indicate useful frequency, and a lower amplitude is not necessarily useless. More importantly, Figure 1 shows that a Fourier basis function is time-dependent when the input length is not divisible by a certain frequency level. Consequently, the mapping in the frequency space is not enough and fails to capture time-frequency relationships.
>
> We provide a new perspective that real and imaginary parts can be interpreted as the coefficients of cosine and sine basis functions at different granularities of frequencies. However, existing Fourier-based methods do not involve such basis functions, thus failing to interpret these coefficients correctly, as shown in Figure 1. Accordingly, we propose the Fourier Basis Mapping (FBM) by incorporating basis functions to extract more efficient time-frequency features to solve two inconsistency issues, making the downstream mapping much easier. With time-frequency features, very simple mapping networks (L-vanilla linear network and NL-three-layer MLP) can achieve the SOTA in any circumstances. We evaluate our insights through three FBM variants against four categories of LTSF methods: (1) Linear method: NLinear; (2) Transformer-based methods: FEDformer, BasisFormer, iTransformer, Pathformer, and PatchTST; (3) Fourier-based methods: FEDformer, FreTS, N-BEATS, CrossGNN, TimesNet, and FiLM; and (4) MLP-based methods: N-BEATS, FreTS, and TimeMixer. Both FBM-L and FBM-NL achieves the SOTA performance of LTSF on eight real-world datasets or of STSF on datasets PEMS.
>
> We will further revise the paper to thoroughly clarify your raised concerns.
>
>
> **Response to your point (3)**
>
> We thank you for your careful reading, and we will fix these spelling inconsistencies in the final version.
>
> Please be noted, FBM is our method.
>
>
> **Concluding remarks**
>
> As you would notice from our intensive responses to other reviewers, during this tight rebuttal period, we have been working day and night and making our unreserved efforts to address all comments and suggestions. Thanks to the constructive comments of all reviewers including you, this rebuttal has substantially improved our work quality, with major revisions and improvements including:
> 1)	Clarifying the motivation and insights of our method, in particular, redrawing Figure 1 in the PDF file, and rewriting the introduction, as shown in the response to Reviewer Z6Ax;
> 2)	Adding new baseline methods including Pathformer, iTransformer, TimeMixer, and TimesNet;
> 3)	Adding new datasets, including PEMS04 and PEMS08.
>
> **As Reviewer Z6Ax concluded, and the above added experiments have shown, our work discloses a new perspective that can outperform the SOTA baselines with very simple models,** which is quite valuable for machine-efficient learning of complex time series.
>
> We appreciate that both Reviewers Z6Ax and akVH acknowledge our responses. We understand you substantially downgraded your score based on your above three points. We hope our intensive clarifications in this response have clarified your comments, and we do appreciate your cross-reference with other reviewer’s comments and our responses to their comments.
>
> You are more than welcome to verify our codes as well.
> We appreciate your valuable time on reading this lengthy response, and any further comments or suggestions to further improve our work. Many thanks.

---

### Official Review · Reviewer_akVH · 2024-07-11

**Soundness:** 2
**Presentation:** 3
**Contribution:** 2
**Rating:** 6
**Confidence:** 4

**Summary:**

This work rethinks the discrete Fourier transform from a basis functions perspective, identifying two key issues in existing Fourier-based methods: inconsistent starting cycles and series lengths. To address these, the paper proposes the Fourier basis mapping model (FBM), which leverages Fourier basis expansion to obtain a mixture of time and frequency features. Experiments show FBM outperforms diverse baselines on long-term forecasting tasks, with the linear FBM-L performing better for noisy data and nonlinear FBM-NL/NP better for less noisy data, highlighting the importance of combining time and frequency information.

**Strengths:**

S1: The Fourier basis mapping (FBM) model effectively combines time and frequency domain features, outperforming a diverse set of baselines including linear, Transformer, and other Fourier-enhanced methods on long-term time series forecasting tasks.

S2: The different FBM variants (linear and nonlinear) can adaptively select the appropriate model based on the noise characteristics of the data, demonstrating the importance of the time-frequency mixed features.

**Weaknesses:**

W1: The method is primarily validated on long-term time series forecasting tasks, and its applicability to short-term forecasting tasks requires further exploration.

W2: While the FBM model exhibits better interpretability compared to more complex models like Transformers, further analysis of its internal mechanisms is needed to better understand the role of the time-frequency mixed features.

W3: There is a lack of visual showcase demonstrations, making it difficult to provide a qualitative evaluation of the experiments. Additionally, there is a lack of efficiency analysis regarding the computational complexity.

W4: More comparisons with state-of-the-art models are missing, which is crucial for evaluating the quality of the work.

**Questions:**

Q1: How does your proposed FBM model perform in short-term forecasting tasks, especially in comparison to its performance in long-term time series prediction? Are there significant differences? Do you have plans to explore the applicability of FBM in short-term forecasting tasks in future research, such as in evaluations on the M4 dataset and PEMS dataset?

Q2: You mentioned that FBM has better interpretability compared to complex Transformer models. Could you further analyze the internal mechanism of FBM and elaborate on how the spatiotemporal mixed features affect the model's predictive performance? This is crucial for a better understanding of the advantages of FBM.

Q3: It would be helpful to provide more analysis on model parameter sensitivity, efficiency analysis of model computational complexity (including runtime and memory overhead), detailed settings of hyperparameters, and visual showcases.

Q4: Could more comparative experiments with state-of-the-art models such as N-Hits[1], iTransformer[1], TimeMixer[2], and TimesNet[3] be provided? This is crucial for a comprehensive evaluation of this work.

[1]N-Hits: Neural Hierarchical Interpolation for Time Series Forecasting

[2]iTransformer: Inverted Transformers Are Effective for Time Series Forecasting

[3]TimeMixer: Decomposable Multiscale Mixing for Time Series Forecasting

[4]TimesNet: TimesNet: Temporal 2D-Variation Modeling for General Time Series Analysis

**Limitations:**

The authors have not addressed limitation of their work. I would suggest the authors to include such section in the final version of their work.

---

> ### Author Rebuttal · Authors · 2024-08-07
>
> We thank you for your constructive suggestions, which will be fully reflected in the final version. Please refer to the new experiment results in the one-page PDF attached to the global response.
>
> **Weakness W1 and Question Q1**
>
> Table 3 in the PDF shows new short time-series forecasting (STSF) results on the PEMS dataset, but time constrains completing experiments on the M4 dataset (LTSF methods including Pathformer, iTransformer, CrossGNN, FiLM, PatchTST, NLinear and FreTS do not test on M4). FBM variants continuously achieve the SOTA performance for STSF on the PEMS datasets.
>
>
> **Weakness W2 and Question Q2**
>
> Yes, the main contribution of our work is on extracting more interpretable time-frequency features enabling more effective downstream mapping. Consequently, we can use very simple mapping networks (L-vanilla linear network and NL-three-layer MLP) to achieve the SOTA.
>
> The internal mechanisms of FBM variants are provided in Figure 6 in Appendix. Assume the look-back window $T=336$ and forecast horizon $L=96$, FBM-L decomposes a time series into $\frac{T}{2}+1$ pieces at different frequency granularities by Eq. (3). Thus, our FBM-L mapping uses a “nn.Linear (336 $\times$ 169, 96)”, and FBM-NL includes two additional nn.Linear layers with activation functions.
>
> 1.	Performance: Our ablation study evaluates the comparisons between FBM-L vs NLinear and FBM-NP vs PatchTST in Table 1 and FBM-NL vs TimeMixer in Table 2 in the attached PDF, showing the effectiveness of the time-frequency features. The first compares two linear networks, the second compares two PatchTST networks, and the last compares two MLP networks. FBM-L outperforms NLinear on all datasets and forecast horizons. The average MSE and MAE of NLinear are 0.3135 and 0.3395, respectively, which drops to 0.3034 and 0.3297 for FBM-L. For PatchTST, the average MSE and MAE are 0.3079 and 0.3360, respectively, which drops to 0.3058 and 0.3340 for FBM-NL. FBM-NP performs better than PatchTST in most cases. The improvement on PatchTST is less significant because PatchTST primarily considers the time domain. It is worth mentioning that TimeMixer's structure and hyperparameters are well-tuned for each dataset across different forecast horizons, but FBM-NL uses the same structure and hyperparameters all the time for better reproducibility. However, FBM-NL still performs better than TimeMixer in most cases (6 out of 8). With our time-frequency features, our model achieves SOTA performance on all datasets for both LTSF and STSF simply with one or three layers even without tuning.
>
> 2.	Interpretability: We empirically explain why the time-frequency features are better. The previous frequency mapping suffers from issues of inconsistent starting cycles and series lengths. This is because crucial information is stored in the amplitude and phase of the cycle, validated in Eq. (3) and (4). Additionally, a Fourier basis function is time-dependent when the input length is not divisible by the frequency level (see Figure 1 in the PDF). The mapping in frequency space cannot capture time-frequency relationships. Thus, Section 5.3 visualizes how FBM-L considers time-frequency relationships. Since the datasets Electricity and Traffic are more stable than ETTh1 and ETTh2, the weights for the former datasets are closer to the Fourier basis than those for the latter datasets. This implies that FBM-L considers more time-frequency relationships in ETTh1 and ETTh2 than in Electricity and Traffic, leading to more significant improvement.
>
> **Weakness W3 and Question Q3**
>
> We revise the paper for better visual showcase. FBM-L and FBM-NL are listed as two independent columns in Table 1, with a new baseline TimeMixer added.
>
> The efficiency analysis is conducted on ETTh1 with the same setting, here is the training speed for one epoch:
>
> |Model|FBM-L|FBM-NL|FBM-NP|NLinear|PatchTST|N-BEATS|CrossGNN|FEDformer|FreTS|FiLM|BasisFormer|TimeMixer|Pathformer| iTransformer|TimesNet|
> |-|-|-|-|-|-|-|-|-|-|-|-|-|-|-|-|
> |Time|4.86|10.7|34.21|1.36|33.30|5.21|23.50|161.29|28.69|148.50|6.75|10.91|241.23|11.64|12.18|
>
> FBM-L and FBM-NL rank second and fifth in terms of training speed, respectively, because they just need to train one and three nn.Linear layers, respectively. The memory is $\mathbf{O}(L^2/2)$, but one can easily combine some frequency levels to reduce it. For FBM-NL, we use the same hyperparameters for each dataset across different horizons for better reproducibility, see Figure 6 in Appendix. FBM-NP uses the same optimal hyperparameters as PatchTST.
>
>
> **Weakness W4 and Question W4**
>
> Table 2 in the PDF shows the experiment results of new baselines iTransformer, TimeMixer, TimesNet, and Pathformer [1]. Due to time constraints, we cannot finish experiments on N-Hits. However, we can refer to experiment results on the same dataset reported in their paper for comparison. For example, they report 0.401 MSE and 0.413 MAE on ETTm2 (L=720), our FBM-L achieves 0.364 MSE and 0.381 MAE. Table 2 shows our FBM variants continuously achieve the SOTA performance against these new baselines.
>
> **Limitation**
>
> For the limitation part, there may be some misunderstanding. Our method does consider the monotonic trend effect, as the Fourier basis has time dependent and independent frequency basis, which helps separate the trend and seasonal effects like DLinear. However, how to use the previous trend to predict future trends is still a major challenge, which hasn’t been solved for every method, as real-world data usually don’t have a patternable trend effect.
>
> Ref:
>
> [1] Pathformer: Multi-scale Transformers with Adaptive Pathways for Time Series Forecasting
>
> We have thoroughly addressed all comments and produced a new version of the paper, thank you for any other suggestions.

---

> > ### Comment · Reviewer_akVH · 2024-08-08
> >
> > Thank you for the author's response. I have carefully read your reply, but my concerns have not been alleviated. I still have the following questions:
> >
> > Q1. Why are only the ETTh1 and ETTh2 datasets included in the new baseline comparison experiments? Additionally, why do the reported results of the new baselines differ significantly from those in their paper? If the experimental settings were modified, please describe the parameter settings for each model in detail.
> >
> > Q2. The analysis of model performance should encompass running time, model parameter size, and GPU memory usage. The current analysis is too simplistic, and it should also examine how the model's efficiency changes with increasing input length.
> >
> > Q3. Moreover, a sensitivity analysis of model parameters, detailed hyperparameter settings, visual showcases, and error bars with confidence intervals for prediction results should be provided.
> >
> > Q4. In addition to empirical analysis, is there a more intuitive visual analysis of the model's interpretability? This is crucial for enhancing our understanding of the model.
> >
> > Q5. Furthermore, many works on time-frequency features in time series, such as FITS. What are the differences between the proposed model and those? Using Fourier transforms to process time series is a long-established topic; what are the novel contributions of our approach?
> >
> > Resolving the above issues is crucial for improving the quality of this work. I will carefully consider the author's responses before making a final decision.

---

> ### Comment · Reviewer_Z6Ax · 2024-08-07
>
> I have similar concerns.
>
> W3: There is a lack of visual showcase demonstrations, making it difficult to provide a qualitative evaluation of the experiments.
>
> Q2: Additionally, there is a lack of efficiency analysis regarding the computational complexity.
>
> You say that "The internal mechanisms of FBM variants are provided in Figure 6 ", but I mean that could you please provide some illustration to show how the proposed method can improve the performance? I need an intuitive explanation.
>
> I know that you can no longer provide a Figure in the discussion phase. Please tell me your plan to improve the presentation.

---

> ### Comment · Reviewer_Z6Ax · 2024-08-07
>
> I think the author's response is worth consideration. This is a cool paper because very simple mapping networks (L-vanilla linear network and NL-three-layer MLP) can achieve the SOTA in any circumstance with the proposed time-frequency features.

---

> > ### Comment · Reviewer_akVH · 2024-08-08
> >
> > Thank your suggestion. Similar to your perspective, I recognize the potential in the paper; however, I continue to have several inquiries. The author's responses have not fully addressed my concerns. I will take the time to thoroughly evaluate the author's forthcoming replies before concluding.

---

> ### Author Response · Authors · 2024-08-08
> **Intuitive explanation to the FBM model's performance**
>
> First, we thank you for checking the comments providing feedback to our rebuttal to **Reviewer akVH**.
>
> Regarding the **Weakness W3** and **Question Q2**, we have responded to them, please find the corresponding responses to **Weakness W3** and **Question Q3** in the rebuttal to **Reviewer akVH**.
>
> Further, regarding the internal mechanism of our FBM, we provide the following intuitive explanation.
>
> The effectiveness of the time-frequency feature lies on that it shares the similarity with methods like DLinear and Autoformer, where a model's performance can sometimes be improved using a moving average to separate trend and seasonal effects. However, this approach is not always effective because it requires determining an appropriate kernel size for the moving average. In our method, we use the Fourier basis to decompose all potential effects into pieces (e.g., 2-hour cycles, 24-hour cycles (daily), 168-hour cycles (weekly), and k-hour cycles) so that different effects can fall into different frequency levels. Then, FBM can consider all the potential effects hierarchically, making downstream mapping much easier. Additionally, the Fourier basis includes time-dependent and time-independent basis functions, which further help automatically separate trend and seasonal effects. The mapping allows the model to consider all the potential effects interactively and hierarchically and remove noises in both time and frequency spaces. You can think of this as an upgraded version of DLinear. While DLinear only performs better than NLinear in very small cases, our FBM-L consistently outperforms NLinear for all the time. In conclusion, FBM separates all potential effects and allows the neural network to consider those effects automatically and interactively, while DLinear only considers the trend and seasonal effects and whether it is a good separation largely depends on the kernel size you choose.

---

> ### Author Response · Authors · 2024-08-10
> **Further Responses to Your New Questions Q1 and Q5**
>
> Thank you for reading our responses and raising further questions. Below, we respond to each of your questions.
>
> **Q1**
>
> The ETTh1 and ETTh2 datasets are used for two reasons. First, they are widely used for LTSF, appearing in almost all relevant methods as cited in their experiments. The second reason is that the experiment settings on this data are always the same except for the input sequence length, making the results reported in their papers comparable with each other.
>
> Accordingly, in our global response to all reviewers, we point out that all the experiment results are conducted with an input sequence length of 336, rather than 96, except for Pathformer, as its hyperparameters do not work with 336. This is consistent with the settings in our paper. A length of 336 always produces better results than 96. This is also evidenced by the following papers, where Group A (with sequence length 336) consistently reports lower MSE and MAE than Group B (sequence length 96).
>
> Group A:
>
> 1.	PatchTST: A Time Series is Worth 64 Words: Long-term Forecasting with Transformers
> 2.	NLinear: Are Transformers Effective for Time Series Forecasting
>
> Group B:
>
> 1.	Pathformer: Multi-scale Transformers with Adaptive Pathways for Time Series Forecasting
> 2.	iTransformer: Inverted Transformers Are Effective for Time Series Forecasting
> 3.	TimeMixer: Decomposable Multiscale Mixing for Time Series Forecasting
> 4.	TimesNet: Temporal 2D-Variation Modeling for General Time Series Analysis
>
> Therefore, we set the input sequence length to 336. Thus, our experiment settings are the same as that of PatchTST and NLinear, but slightly different from Pathformer, iTransformer, TimeMixer, and TimesNet, as they use the input sequence length of 96. As the same hyperparameters in their official codes are used to reproduce their results, our reported performance DOES NOT differ significantly from those appearing in their papers. In fact, our reported MSE and MAE scores are close to theirs with an acceptable difference, which is very common in reproducing existing methods. In fact, if you compare the MSE and MAE scores reported in PacthTST with the scores of those papers in Group B reported for PatchTST, you would find a more significant difference. In addition, as TimesNet uses 0 and 1 normalization, it is not directly comparable.
>
> **Q5**
>
> Let’s answer your question Q5 firstly before addressing others, as this is the most important question.
>
> As suggested by Reviewer Z6Ax in the first round of review, we generated a new **Figure 1 shown in the PDF for global response** to all reviewers. The new figure better illustrates, compares, and summarizes existing Fourier-based methods with our FBM and emphasizes our main contributions. It also shows visual use cases to illustrate their differences.
>
> You may misread/misunderstand the motivation and main contributions of our work. Please let us explain below. In fact, we revised the last two paragraphs of the introduction to clarify the contributions, requested by Reviewer Z6Ax, who has adjusted his/her misreading.
>
> Please refer to the response titled **Here is the revised last two paragraphs of Introduction** to Reviewer Z6Ax
>
> Thus, our FBM has a new insight into deep time series forecasting using the basis functions with time-frequency features, making the following main differences:
>
> 1.	Existing Fourier-based mapping is conducted in the frequency space, while FBM is conducted in the time-frequency space.
> 2.	Existing methods face issues of inconsistent starting cycles and inconsistent series lengths, but addressed by FBM.
> 3.	Existing methods ignore that Fourier basis could be time dependent, but addressed by FBM.
>
> We find that FITS you suggested is a very interesting and relevant model to compare, so we’ll cite this paper in the related work and take it as a new baseline to update results in Table 1. FITS also addresses the issue of inconsistent starting cycles but does not resolve the issue of inconsistent series lengths and fails to consider time-frequency relationships, as its mapping is conducted in the frequency space. However, the existing mapping in the frequency space completely ignores the time dependent frequency information. This is because a Fourier basis function is time-dependent when the input length is not divisible by the frequency level (**See the time-dependent basis in Figure 1 of the PDF**). FITS primarily focuses on reducing the size of model parameters. In contrast, the main contributions of our paper lie in disclosing the issues of inconsistent starting cycles and inconsistent series lengths with existing Fourier-based methods and demonstrating the effectiveness of the time-frequency features. With time-frequency features, remarkably simple mapping networks (L-vanilla linear network and NL-three-layer MLP) can achieve the SOTA in any circumstance.
>
> Pls read the following responses to Q2-Q4 as well.

---

> ### Author Response · Authors · 2024-08-10
> **Further Responses to Your New Questions Q2 and Q3**
>
> **Q2**
>
> FITS reports their results w.r.t. the Parameters and MACs, we follow this to compare FBM with it for your convenience as our model’s parameters and memory are calculable by hand. The internal mechanisms of FBM variants with their hyperparameters are provided in Figure 6 in Appendix.
>
> Assuming $T=96, L=720$, the same settings as FITS, the number of parameters of FBM-L is (96$\times$49) $\times$720=3.38M.
>
> FBM-NL has three layers, totalling (96$\times$49) $\times$1440+1440$\times$1440+1440$\times$720=9.88M. The MACs of FBM-L and FBM-NL are 49 and 96 times larger than that of NLinear, respectively. The hidden states in the second and third layer of FBM-NL are fixed, which is 1440, see Figure 6 in Appendix. FBM-NL uses the same structure and hyperparameters all the time for better reproducibility.
>
> We also use the hyperparameters of FBM-NP as the optimal hyperparameters for PatchTST. Thus, the only difference lies in the first initial projection layer, and all the downstream structures are completely the same. In PatchTST, the initial projection layer is nn.Linear($\frac{2T}{P}$, K), while ours is nn.Linear($\frac{T^2}{2P}$, K). The optimal P and K of PatchTST for data Electricity is 16 and 512 respectively, reported in their official codes. Thus, the difference is $(\frac{96}{2}-2)\times512 \times \frac{96}{16}$=0.14M. FBM-NP has 0.14M more parameters, but the same MACs as PatchTST.
>
> The following compares our FBM with the results in Table 3 in FITS:
>
> Model|Parameter|MACs
> -|-|-
> TimesNet|301.7M| 1226G
> FiLM|14.91M |5.97G
> FEDformer|20.68M|4.41G
> PatchTST|1.5M|5.07G
> DLinear|0.14M|0.04G
> NLinear|0.07M|0.02G
> FBM-L|3.38M|0.96G
> FBM-NL|9.88M|1.82G |
> FBM-NP|1.64M|5.07G
>
> FBM has lower MACs than every transformer-based models. The number of total parameters is also lower than most Transformer-based methods. It is worth mentioning that FBM-NP improves the performance of PatchTST without substantially increasing its complexity, using the same optimal hyperparameter as PatchTST without tuning, which further demonstrates the effectiveness of time-frequency features. If the sequence length increases, the number of parameters and MACs will increase quadratically for FBM-L and increase quadratically in the first layer but remain the same for the rest two layers of FBM-NL. However, all transformer-based methods suffer from a quadratic increase, thus our FBM variants always have lower MACs compared with transformer-based models.
>
> In addition, FBM-L and FBM-NL rank second and fifth in terms of training speed, respectively, with T=336 and L=336. This is not a large language model, the training speeds and MACs play a more significant role rather than model parameters, which demonstrates that FBM variants are efficient comparing to existing LTSF methods.
>
> We hope the above explanation helps you capture the internal mechanisms of FBM and address your concern about efficiency. We’ll reflect the above efficiency analysis in the appendix of the final paper, due to limitation of space and time in rebuttal.
>
> **Q3**
>
> The internal mechanisms of FBM variants are provided in Figure 6 in Appendix with all the hyperparameters and layers provided, making the model’s parameters calculable by hand, as shown in the response to your question Q2.
>
> The hyperparameters for FBM-NL are fixed, the same as those used in PatchTST for FBM-NP, to ensure better reproducibility. While a sensitivity analysis is not meaningful here, we address your concern by analyzing the hidden states from 720 to 1440 for FBM-NL on ETTh1:
>
> FBM-NL|hidden=1440|hidden=1440|hidden=720|hidden=720
> -|-|-|-|-
> Error|MSE| MAE| MSE| MAE|
> ETTh1-L=96|0.368|0.395|0.370|0.397
> ETTh1-L=192|0.408|0.418|0.409|0.420
> ETTh1-L=336|0.425|0.430|0.427|0.433
> ETTh1-L=720|0.456|0.466|0.462|0.470
>
> The performance of varying hidden states can differ across datasets, but tuning the hyperparameters will undermine reproducibility with changing settings (e.g., sequence length). Therefore, we have included a sensitivity analysis for sequence lengths of 96 and 336 in Table 4 of Appendix, which shows that a sequence length of 336 is more robust and produces better results for most baseline methods.
>
> The MSE and MAE results of FBM variants in the tables are calculated by averaging the results of two runs. Since we didn’t store the initial error bar in the previous experiments, we cannot provide the full reports here due to limit of space and time, but we will add the error bar analysis of FBM variants to update the Appendix. Here is the error bar analysis conducted on ETTh1:
>
> Model| FBM-L|FBM-L|FBM-NL|FBM-NL|FBM-NP|FBM-NP
> -|-|-|-|-|-|-
> Error|MSE| MAE|MSE|MAE|MSE|MAE
> ETTh1-L=96|0.367±0.01| 0.391±0.01 | 0.369±0.01| 0.395±0.01 | 0.367±0.01|0.394±0.01
> ETTh1-L=192| 0.403±0.01 |0.410±0.01 | 0.408±0.01 | 0.418 ±0.01|0.407±0.01|0.417±0.01
> ETTh1-L=336|0.420±0.03| 0.420±0.01 | 0.428±0.04| 0.432 ±0.03 |0.429±0.05|0.435±0.05
> ETTh1-L=720|0.416±0.03|0.439±0.01 | 0.456±0.02 |0.466±0.02 | 0.442±0.04 |0.462±0.03

---

> ### Author Response · Authors · 2024-08-10
> **Further Responses to Your New Questions Q4**
>
> **Q4**
>
> In fact, we have provided empirical visual analysis of the time-frequency features:
>
> 1. The updated Figure 1 in the PDF with the global response to all reviewers visualizes the Fourier basis expansion, where Fourier basis can be categorized into time dependent basis and time independent basis.
>
> 2. Section 5.3 visualizes the weight of FBM-L to explain how FBM-L considers the time-frequency relationships.
>
> First, we find that a Fourier basis function is time-dependent when its input length is not divisible by the frequency level (see time dependent basis in Figure 1 in the PDF). Mapping in the frequency space cannot capture the time-frequency relationships.
>
> Second, Section 5.3 visualizes how FBM-L considers the time-frequency relationships, addressing the interpretability of FBM in the response to **Weakness W2**.
>
> We thank you for the detailed comments and hope we have clarified your concerns. We will reflect these responses in the final version. Please let us know if you have any other constructive suggestions. Many thanks.

---

> > ### Comment · Reviewer_akVH · 2024-08-11
> >
> > I would like to express my gratitude to the author for the comprehensive response, which has addressed several of my initial concerns. Consequently, I have increased my assessment score. I strongly encourage the author to incorporate this information into the final version of the manuscript. Nevertheless, I still have some inquiries. Could the author provide a comprehensive comparison of the ETT dataset alongside the Traffic, Electricity, and Weather datasets in relation to all the newly introduced baselines? I anticipate the author's further elaboration on this matter.

---

> ### Author Response · Authors · 2024-08-12
> **Further comparison**
>
> **Further comparison**
>
> Thank you for reading our responses carefully and positive acknowledgment of our responses. Definitely, the changes will be reflected in the final version. Below, we respond to your new enquiry.
>
> Due to time constraints, we have not conducted experiments on TimesNet, as it is not the latest model. Below are the results of the newly introduced baselines with FBM-NL and FBM-NP on the Traffic, Electricity, and Weather datasets.
>
> Model | FBM-NL | FBM-NL | FBM -NP | FBM-NP |iTransformer|iTransformer|TimeMixer|TimeMixer|Pathformer|Pathformer
> -|-|-|-|-|-|-|-|-|-|-
> Error| MSE | MAE| MSE | MAE|MSE|MAE| MSE|MAE|MSE|MAE
> Electricity L=96| 0.106 | 0.199 | 0.107 | 0.199 | 0.109 | 0.205 | 0.108 | 0.200 | 0.113 | 0.199
> Electricity L=192| 0.122 | 0.215 | 0.121 | 0.210 | 0.125 | 0.222 | 0.122 | 0.212 | 0.127 | 0.211
> Electricity L=336| 0.141 | 0.231 | 0.140 | 0.229 | 0.141 | 0.236 | 0.143 | 0.233 | 0.147 | 0.231
> Electricity L=720| 0.175 | 0.265 | 0.175 | 0.264 | 0.163 | 0.262 | 0.183 | 0.270 | 0.176 | 0.260
> Traffic L=96| 0.283 | 0.227 | 0.288 | 0.231 | 0.292 | 0.244 | 0.290 | 0.240 | 0.362 | 0.268
> Traffic L=192| 0.289 | 0.231 | 0.293 | 0.234 | 0.301 | 0.246 | 0.304 | 0.350 | 0.371 | 0.263
> Traffic L=336| 0.293 | 0.236 | 0.298 | 0.239 | 0.306 | 0.250 | 0.313 | 0.254 | 0.374 | 0.267
> Traffic L=720| 0.306 | 0.247 | 0.309 | 0.248 | 0.316 | 0.264 | 0.320 | 0.267 | 0.384 | 0.290
> Weather L=96| 0.152 | 0.200 | 0.156 | 0.204 | 0.162 | 0.211 | 0.158 | 0.204 | 0.151 | 0.191
> Weather L=192| 0.194 | 0.242 | 0.198 | 0.245 | 0.204 | 0.249 | 0.197 | 0.246 | 0.202 | 0.242
> Weather L=336| 0.244 | 0.282 | 0.248 | 0.285 | 0.248 | 0.285 | 0.242 | 0.281 | 0.260 | 0.283
> Weather L=720| 0.317 | 0.334 | 0.319 | 0.337 | 0.322 | 0.335 | 0.319 | 0.335 | 0.337 | 0.336
>
> Both FBM-NL and FBM-NP achieve better performance than all newly introduced baseline methods in most cases, except for three: (1) Electricity L=720, where iTransformer achieves the best MSE, and Pathformer achieves the best MAE; and (2) Weather L=336, where TimeMixer achieves the best MSE and MAE; and (3) Weather L=96, where Pathformer achieves the best MSE and MAE. It's worth noting that there is still potential to improve the performance of our FBM variants, as we used the fixed hyperparameters.
>
> We will also reflect these comparisons in the final version. Please let us know if you have any other constructive suggestions, we will continuously try our best to address them and improve the paper quality. Many thanks.

---

> > ### Comment · Reviewer_akVH · 2024-08-12
> >
> > Thanks for your detailed response. After careful examination, I found some issues. The results reported on the Electricity and Traffic datasets differ significantly from those in other papers. What could be the reason for this?

---

> ### Author Response · Authors · 2024-08-12
>
> Thanks for your careful examination.
>
> In line 441 of the paper, we mentioned that we selected the last 20 dimensions of the Traffic and Electricity datasets to accelerate training speed. This decision was made because most baseline methods were originally designed for the sequence length of 96 rather than 336. They did not account for the quadratic increase in both MACs and the approximately fourfold increase of training time when the sequence length is expanded, making them extremely slow to train with a sequence length of 336—approximately ten times slower or even more.
>
> To ensure a fair comparison, we also tune their learning rates, as their experimental settings change with the increase of input sequence length from 96 to 336. Without this adjustment, these baselines would not perform anywhere close to our methods. However, it would be impossible to fully optimize their models, given that training for once could take days or even weeks on these datasets for each baseline. For example, Pathformer takes almost a day to train with the input sequence length of 96 under our settings. Thus, if we used the full dimension in our experiment, we would not be able to complete the training and present you new results before this rebuttal ends. Furthermore, if we expanded the input sequence length to 336 for Pathformer, assuming their hyperparameters would work for 336, it would take a few weeks to train just one baseline model using our computational facilities. It would have been impossible to help them tune the learning rate, making a fair comparison unattainable. Therefore, we slightly modified the experimental settings to preserve the characteristics of the data while ensuring that a fair comparison could be conducted.
>
> We hope this makes sense to you, otherwise we would be continuing our experiments as well even though we can't provide more comparisons before this rebuttal ends. We hope our explanation addresses your concern. If you have any other concerns, please let us know. We will continuously try our best to address them. Many Thanks.

---

> > ### Comment · Reviewer_akVH · 2024-08-12
> >
> > Thank you for your further response. I have tried running the baseline models and datasets you mentioned, and I was able to complete the tests within a limited timeframe. Therefore, I don't understand why certain dimensions must be selectively chosen for acceleration. TimeMixer and iTransformer are relatively efficient models, and there shouldn't be any efficiency issues when testing well-known datasets. Moreover, Reviewer 8ZRG pointed out that the results you reported differ significantly from those in the paper, which may lead to ambiguity. For example, I also attempted to run tests using the code of iTransformer and found that the results were generally consistent with those reported in the paper. Therefore, I don't quite understand why your reported results deviate significantly. Based on these two points, I have decided to maintain my score.

---

> ### Author Response · Authors · 2024-08-12
> **Regarding performance deviation by different lengths and measures**
>
> We thank you for your follow-up of our response and further comments during your busy agenda.
>
> Regarding the performance difference concerning iTransformer on the ETTh1 dataset mentioned by Reviewer 8ZRG, this reviewer actually misread or overlook the different input sequence lengths used by our method (with length 336) vs that by iTransformer (with length 96) on dataset ETTh1, and the different performance measures for ours (per MAE and MAPE) vs iTransformer (per MSE and MAE) on dataset PEMS04. Please find our responses titled **Reponses to your points –1** to **Reponses to your points –3** to Reviewer 8ZRG, where we comprehensively clarified the misreading and justified why different lengths and measures were used.
>
> In addition, please refer to our previous response **Q1** in the second round of response titled **Further Responses to Your New Questions Q1 and Q5** to you. We have thoroughly explained that this discrepancy is due to the different input sequence lengths used in the iTransformer paper and ours, i.e., 336 is applied in our work versus 96 in theirs.
>
> For example, if you compare the MSE and MAE scores reported in PacthTST with the scores reported by iTransformer for PatchTST, you will find even more significant difference. We collect and present their results in the following table:
>
> PatchTST’s results | In PatchTST's paper with T=336 | In PatchTST 's paper with T=336 | In iTransformer's paper with T=96 | In iTransformer's paper with T=96
> -|-|-|-|-
> Error|MSE| MAE| MSE| MAE|
> ETTh1 L= 96|0.375|0.399|0.414|0.419
> ETTh1 L=192|0.414|0.421|0.460|0.445
> ETTh1 L=336|0.431|0.436|0.501|0.466
> ETTh1 L=720|0.449|0.466|0.500|0.488
>
> Second, Reviewer 8ZRG overlooked the different performance measures used in evaluating iTransformer on the PEMS04 dataset: we report the MAE and MAPE results, as MAE and MAPE are the most reliable metrics for the evaluation on the PEMS04 dataset. In contrast, iTransformer reports the MSE and MAE results, which are less appropriate for that dataset. In fact, our experiments report a lower MAE score than that reported in the iTransformer paper because we use the input sequence length of 336 rather than 96. This result further demonstrates that the input sequence length of 336 is more robust.
>
> Regarding the above different evaluation settings and results, you are more than welcome to test our shared codes in the Supplementary Material (mentioned in line 524 in our paper  and our global response to all reviewers) to verify our settings, evaluation measures and results. Our codes follow the same structure as iTransformer, so it is easy to run if you are familiar with it. Once you have chance to test our codes, you would verify our above clarification and Reviewer 8ZRG’s misreading as well. You could verify FBM-L on the ETTh1 and ETTh2 datasets first, as it runs very quickly.
>
> Finally, while we acknowledge your point that TimeMixer and iTransformer are efficient, other baseline methods, such as Fedformer, Pathformer, and FiLM, do not achieve the same level of efficiency. This is precisely why most papers only verify their methods using the input sequence length of 96 rather than 336, even though 336 has been demonstrated by NLinear and PatchTST to be a better choice. While the ETTh1 and ETTh2 datasets are quick to train, the training on the Electricity and Traffic datasets is much slower, especially since most baseline methods significantly increase the dimensionality of hidden states on those datasets.
>
> We sincerely hope our further explanations have addressed all of your concerns relating to Reviewer 8ZRG’s feedback as well as your previous concerns. If you have any trouble in verifying our codes, finding any issues there, or having any further comments, please feel free to let us know. We are endeavoured to address them as much as we can. Many thanks.

---

> > ### Comment · Reviewer_akVH · 2024-08-13
> >
> > Thanks again for your detailed response. I have read all the feedback and discussions, and I executed your code. The results of ETT met expectations, so I believe this work has potential. However, just as Reviewer 8ZRG, I think there is still significant room for improvement:
> >
> > 1. I strongly recommend that the authors clearly describe the experimental settings and implementation details. For example, selecting different dimensions from the dataset for evaluation leads to results that differ significantly from other papers, which can easily cause misunderstandings and ambiguities.
> >
> > 2. I suggest standardizing all experimental settings for the experiments. If you are using an input length of 336, I recommend reporting the results of other baselines under unified parameter settings. Additionally, parameters such as learning rate and epoch should also be kept within the same range to ensure a fair comparison.
> >
> > 3. Furthermore, I hope the authors can provide more visual examples to illustrate the uniqueness of your proposed method in time-frequency modeling. Currently, there are many similar works, but this has not been addressed in your paper.
> >
> > In summary, I hope the authors can further improve their work for better achievements.

---

> > > ### Author Response · Authors · 2024-08-13
> > > **Reply for further improvements**
> > >
> > > We thank you for taking your valuable time to read all the feedback and discussions, and we truly appreciate that you see the potential in our work. During the rebuttal process, we have done our best to address all the comments and continuously improve the quality of the paper. I will address each of your three suggestions individually to further enhance the quality of the paper.
> > >
> > > **Suggestion 1**
> > >
> > > We have included all the experimental settings in Section A.1 of the Appendix and visualized the implementation details of the internal mapping mechanism in Figure 6 of the Appendix. To further enhance readability, we will generate a table to better illustrate all the settings. We acknowledge that one experiment setting has caused misunderstandings. Therefore, in the final version, we will evaluate the full dimensions of the Electricity and Traffic datasets by replacing less efficient baseline methods, such as FEDformer and FiLM, with state-of-the-art methods, including TimeMixer, iTransformer, Pathformer, and FITS.
> > >
> > > **Suggestion 2**
> > >
> > > We have used the optimal hyperparameters reported in the official codes to implement all the baseline methods, ensuring a fair comparison by training them with the same number of epochs. However, we challenge the suggestion to use the similar learning rate for each baseline method. Each baseline model requires a different learning rate, and there is no universal rule for determining whether a larger or smaller learning rate is better for a specific model. For example, our FBM-L model decomposes the time series into pieces, which requires a smaller learning rate.
> > >
> > > Since changing the input sequence length from 96 to 336 can affect the optimal learning rate for each baseline method, we decided to report the optimal learning rate we identified for each method in a table, specifying our experimental settings. This will benefit future research and enhance the reproducibility of the baseline methods.
> > >
> > > **Suggestion 3**
> > >
> > > We have included a new Figure 1 in the global response PDF to illustrate the uniqueness of our proposed method in time-frequency modeling. Figure 1 demonstrates that the uniqueness of our method lies in the inclusion of the basis function, specifically the Fourier Basis Expansion, which helps extract more efficient time-frequency features, thereby making the downstream mapping much easier. In Section 4, we have also provided additional explanations, such as:
> > >
> > > In Line 163: “The process is analogous to decomposing the original time series into $\frac{T}{2}+1$ components with tiered frequency levels, which allows the model to separate various effects hierarchically but identify the affiliated noises.”
> > >
> > > In line 174: ”if we sum along the $\frac{T}{2}+1$ frequency domain, we can obtain the original time series without corrupting the time domain information.”
> > >
> > > These are all unique advantages of our method, and we will add more explanations to the paper.
> > >
> > > Finally, we sincerely thank you and the other reviewers for your valuable comments and suggestions. We have also done our best to improve the quality of our work based on the feedback.

---

> > > > ### Comment · Reviewer_akVH · 2024-08-13
> > > >
> > > > Thank you for your timely and detailed response. I have indeed seen the authors' efforts to improve their paper's quality; in light of this, I have updated my score accordingly. Regarding suggestion 2, I still strongly recommend that the authors use a uniform input length for testing across different models, for example, by using an input length of 336 for all models. Additionally, concerning the learning rate, as previously mentioned, I suggest that the authors test within a unified range (for example, from 0.001 to 0.01), and that the training epochs be tested within a unified range. This is crucial for fair comparisons between experiments; reasonable and fair experimental settings are necessary to obtain valuable comparative results. Finally, I look forward to witnessing the improvements and advancements in your work.

---

> > > > > ### Author Response · Authors · 2024-08-13
> > > > > **Thanks for your positive feedback and your valuable suggestions**
> > > > >
> > > > > Thank you for your prompt feedback, and positive acknowledgement of our efforts.
> > > > >
> > > > > Firstly, we will use a uniform input length of 336 for all baseline methods in the final version and work out the appropriate hyperparameters for Pathformer. Secondly, we will establish a consistent range for the learning rate to ensure that the training epochs for each baseline method are tested within a unified range. Thirdly, we will enhance the presentation of the paper to help readers better understand the motivation and workflow of our method. Lastly, we will include a limitations section to discuss the shortcomings of the current work and potential directions for future improvements. All your valuable suggestions will be reflected in the final version of the paper. Many thanks!

---

> > > > ### Comment · Reviewer_akVH · 2024-08-13
> > > >
> > > > Additionally, I suggest that the authors include a limitations section to discuss the shortcomings of the current work and potential directions for future improvements, which could enhance the impact of the study. Furthermore, regarding the contributions in the time-frequency domain, a more intuitive presentation for the readers should be employed, as the current format appears overly simplistic and makes it difficult for readers to grasp the value of the authors' contributions.

---

> > > > ### Comment · Reviewer_akVH · 2024-08-13
> > > >
> > > > There’s one more point I must address. While reviewing your code, I noticed that you only selected the last 20 dimensions of variables for testing on the Electricity and Traffic datasets. This does not align with current benchmark tests. The reasoning for speeding up training does not hold water. During the lengthy paper preparation period and a two-week rebuttal period, I cannot think of any valid reason for not conducting a complete test. This is a critical flaw, and I strongly urge the authors to conduct full experiments according to benchmark standards; otherwise, the value of this work will be significantly diminished.

---

> > > > > ### Author Response · Authors · 2024-08-13
> > > > > **Response to full evaluation on Electricity and Traffic datasets**
> > > > >
> > > > > Thanks for your valuable suggestions. We will definitely evaluate the full dimensions of the Electricity and Traffic datasets in the final version. We will start to run the code right now and report as much as we can on the Electricity and Traffic datasets.

---

> > > > > ### Author Response · Authors · 2024-08-14
> > > > > **New Further Comparison**
> > > > >
> > > > > Thank you for waiting. We have utilized all available GPU resources to run the experiments, but due to time constraints, we can only report results on the Electricity dataset with some efficient baseline methods at this stage. Pathformer is very slow, even with an input sequence length of 96, and running the Traffic dataset would take approximately three times longer than Electricity dataset. We will include the full set of experiments on the Electricity and Traffic datasets with full dimensions in the final version of the paper. Since FEDformer and FiLM are not efficient, especially with an input sequence length of 336, and do not achieve SOTA performance, they will be replaced with the latest SOTA baseline methods—TimeMixer, iTransformer, Pathformer, and FITS—in the final paper.
> > > > >
> > > > > Model| FBM-NL|FBM-NL|FBM-NP|FBM-NP|PatchTST|PatchTST|iTransformer|iTransformer|TimeMixer|TimeMixer
> > > > > -|-|-|-|-|-|-|-|-|-|-
> > > > > Error|MSE| MAE|MSE|MAE|MSE|MAE|MSE|MAE| MSE|MAE
> > > > > Electricity(ECL) - L=96|   0.132 | 0.227 | 0.133| 0.227 | 0.133   | 0.227    | 0.137   | 0.232  |  0.134  |0.230
> > > > > Electricity(ECL) - L=192| 0.149 |0.243 | 0.149 | 0.242 |  0.151 | 0.244  | 0.156 | 0.249  | 0.153 |0.245
> > > > > Electricity(ECL) - L=336|0.167 | 0.261 | 0.167 | 0.261 |0. 167   |0.261 |  0.171 | 0.266 | 0.172 |0.267
> > > > > Electricity(ECL) - L=720|   0.207 | 0.295 | 0.208 | 0.295 |  0.210   | 0.297   | 0.195   | 0.288  | 0.212 | 0.298
> > > > >
> > > > > Our new results align closely with our previous findings (see the table in our response to you under 'Further Comparison' ) and also show lower MSE and MAE values than those reported in the iTransformer and TimeMixer papers. This further demonstrates that a sequence length of 336 is a more robust setting. You are welcome to compare your results with ours by running the official code for each baseline method and FBM for our codes using the following experimental settings: input sequence length T=336 and a train/validation/test ratio of 6.5/1.5/2.
> > > > >
> > > > >  Many thanks for your valuable suggestions again.

---

### Official Review · Reviewer_Z6Ax · 2024-07-12

**Soundness:** 3
**Presentation:** 1
**Contribution:** 3
**Rating:** 6
**Confidence:** 5

**Summary:**

The paper introduces a novel approach that rethinks the application of the Fourier Transform for time series prediction, proposing a unique Fourier Basis Mapping (FBM) method. It combines both learnable and non-learnable components and demonstrates potential improvements through comprehensive experiments. This paper provides cool solutions to the inconsistent starting cycle and series length issues practically and theoretically. However, significant concerns remain. The paper does not clearly link the identified problems with the proposed solutions, resulting in fragmented logic. Figure 1 and the Introduction section lack clarity and focus. The methods section's reliance on non-learnable structures without clear integration with learnable components raises questions about their effectiveness. The unclear title and purpose of Section 5.3, along with insufficient explanation of denoising capabilities, further weaken the paper. Additionally, the absence of comparative studies and lack of ablation experiments undermine the validation of the proposed method's effectiveness. Addressing these issues could potentially render the paper more acceptable (7 scores).

**Strengths:**

1. The paper presents an innovative approach to time series prediction by utilizing the Fourier Transform (FT), which provides a novel perspective on leveraging frequency domain information for temporal data.

2. The proposed Fourier Basis Mapping (FBM) method effectively integrates both learnable and non-learnable components, showcasing a unique combination that could potentially enhance the feature extraction process and improve prediction accuracy.

3. The extensive experiments conducted demonstrate the potential of the proposed method to improve performance on various time series benchmarks, as evidenced by the quantitative results provided in the study.

**Weaknesses:**

1 The Introduction section inadequately discusses the issues with existing methods. The descriptions are obscure and hard to follow. It is recommended to replace these with more understandable explanations. Two separate paragraphs to elaborate on the identified problems and the corresponding proposed solutions are expected. The current version does not clearly articulate how the method is specifically designed to address each problem, leading to a fragmented logic. Readers may find it difficult to directly correlate the proposed method with the problem-solving benefits claimed.

2 Figure 1 lacks focus and merely replicates descriptions from the text without emphasizing the unique aspects of the proposed method. The distinctive features of the method are not prominently highlighted.

3 The methods section primarily consists of non-learnable structures, which raises significant concerns. The paper does not explain how these non-learnable components integrate with learnable structures to enhance feature extraction, leading to questions about their effectiveness. Therefore,  the title "Effect of Weights for Linear Mapping" in Section 5.3 becomes unclear and feels disjointed, making it difficult to grasp the purpose of this section at a glance. The paper fails to clearly explain why this visualization was conducted and how it supports the stated conclusions. Additionally, the explanation of the model's denoising capabilities is sparse and lacks clarity, seeming somewhat self-serving.

4 A major issue in the experiments is the absence of comparative studies with the same decoder, making it hard to confirm the effectiveness of the proposed frequency domain method. Furthermore, for the key contributions such as denoising and problem-solving, it is suggested to design ablation experiments to validate these effects.

**Questions:**

At first, I want to give this paper an Accept score. However,  I have the following three main concerns after reading this paper:
1. Eq. (3) concludes the main technical contribution towards the long-term time series prediction problem. I agree that we should represent a time series with both the Time and Frequency domains to avoid the inconsistent starting cycle and length issue. However, when I look at the overall framework of the proposed method, I find that the function of this Time-Frequency joint representation is somewhat like extracting features. In this case, I want to know the relationship between the non-parameter feature extractor and the learnable decoder. As your ablation study in Sec 5.3 claims, the decoder has a noise-tuning effect for this module. I doubt this and expect the authors to provide a more comprehensive analysis, or the real effectiveness of the proposed module should be viewed as not fully explored.

2. Experiments are not fair. The comparative experiment should be built with the same architecture. FBM has three variants, and they should be compared with the baseline models separately. The most important two comparisons are FBM-L v.s. NLinear; FBM-NP v.s. PatchTST.  I hope a more fair comparisons can be conducted.

3. The writing should be improved, see Weekness for details.

With the following issues all addressed, this paper can score 7.

**Limitations:**

Experiments are not fair, and the proposed methods are not well elaborated.

---

> ### Author Rebuttal · Authors · 2024-08-07
>
> We thank you for your constructive suggestions, which will be fully reflected in the final version.  Please also refer to the one-page PDF attached and our reply within the global response to all reviewers for the major changes.
>
> **Weakness 1**
>
> We take your kind suggestion and revise the Introduction. As this is important and beneficial for all reviewers to better understand the paper, we have put our answer in **Introduction and Figure 1** within the global response.
>
> **Weakness 2**
>
> We update Figure 1 in the PDF file attached to the response to all reviewers, illustrating, comparing, and summarizing existing Fourier-based methods with FBM and emphasis our main contributions.
>
> **Weakness 3**
>
> You may misread/misunderstand part of our work. Our main contribution is on extracting better initial features enable more effective downstream mapping. With time-frequency features, very simple mapping networks (L-vanilla linear network and NL-three-layer MLP) can achieve the SOTA in any circumstance. The internal mechanisms of FBM variants are provided in Figure 6 in Appendix. Assuming the look-back window $T=336$ and forecast horizon $L=96$, FBM-L decomposes the original time series into $\frac{T}{2}+1$ pieces at different frequency granularities. Thus, our FBM-L mapping uses a “nn.Linear (336 $\times$ 169, 96)”, and the FBM-NL includes two additional nn.Linear layers with activation functions, see Figure 6 in Appendix for details.
>
> The Fourier transform is to decompose a time series hierarchically to remove noise signals. For example, for a two-week hourly time series, $T=336$, with day effects and noises only, after Fourier transform, day effects fall in the frequency levels that are multiples of $14$, and noises fall in all the other frequency levels, making mapping much easier. However, the existing mapping in the frequency space completely ignores the time dependent frequency information and faces two inconsistency issues. This is because a Fourier basis function is time-dependent when the input length is not divisible by the frequency level (see Figure 1 in the PDF). The mapping in the frequency space cannot capture the time-frequency relationships. Thus, Section 5.3 visualizes how FBM-L considers the time-frequency relationships. Since the datasets Electricity and Traffic are more stable than ETTh1 and ETTh2, the weights of the linear layer for the former datasets are closer to the Fourier basis than those for the latter datasets. This implies that FBM-L considers more time-frequency relationships in ETTh1 and ETTh2 than in Electricity and Traffic, leading to more significant improvement. We add a noise attack example to further verify the denoising effect: random Gaussian noises are added to ETTh1 with a 0.1 probability at each input time point during training, resulting in the performance below:
>
> ETTh1|MSE|MAE
> -|-|-
> FBM-L with noise L=192|0. 403|0.411
> FBM-L without noise L=192|0.403|0.411
> FBM-L with noise L=336|0. 418|0.420
> FBM-L without noise L=336|0.418|0.420
>
>
> **Weakness 4**
>
> We add experiments for a fairer comparison of FBM-L, FBM-NL, and FBM-NP in the PDF:
>
> 1.	FBM-L vs NLinear in Table 1
> 2.	FBM-NP vs PatchTST in Table 1
> 3.	FBM-NL vs TimeMixer [1] in Table 2
>
> (1) both with linear networks, (2) both with PatchTST networks, and (3) both with MLP networks.
>
> FBM-L outperforms NLinear on all datasets and forecast horizons. The average MSE and MAE of NLinear are 0.3135 and 0.3395, respectively, which drops to 0.3034 and 0.3297 for FBM-L. For PatchTST, the average MSE and MAE are 0.3079 and 0.3360, respectively, which drops to 0.3058 and 0.3340 for FBM-NL. FBM-NP performs better than PatchTST in most cases. The improvement on PatchTST is less significant because PatchTST primarily considers the time domain. It is worth mentioning that TimeMixer's structure and hyperparameters are well-tuned for each dataset across different forecast horizons, but FBM-NL uses the same structure and hyperparameters all the time for better reproducibility. However, FBM-NL still performs better than TimeMixer in most cases (6 out of 8). These three comparisons demonstrate the effectiveness of the time-frequency features.
>
> Regarding the ablation test in the paper, three layers MLP are sufficient to capture deep time-frequency relationships because FBM-NL always achieves better or the same performance as FBM-NP when non-linear mapping is preferred for a dataset. With our time-frequency features, FBM achieves SOTA performance in almost all circumstances by simply using either a linear layer or a three-layer MLP without tuning.
>
> **Question 1**
>
> FBM considers time-frequency space mapping rather than frequency space mapping. For internal mechanism and empirical thinking, you can refer to our response to Weakness 3. For the effectiveness of time-frequency features, you can refer to our response to Weakness 4.
>
> **Question 2**
>
> Please refer to our response to Weakness 4.
>
> **Question 3**
>
> Please refer to our response to Weakness 1 and the global responses for major changes.
>
> **Limitation**
>
> See our response to Weakness 4 for a fairer comparison and to Weakness 3 for elaborating the proposed methods.
>
> Ref:
>
> [1] TimeMixer: Decomposable Multiscale Mixing for Time Series Forecasting
>
> We have substantially revised the paper addressing all of your comments, please let us know any other kind suggestions.

---

> ### Comment · Reviewer_Z6Ax · 2024-08-07
>
> 1. "We take your kind suggestion and revise the Introduction." Please tell me your revised content.
>
> 2. Please add your clarification "**Our main contribution is on extracting better initial features enable more effective downstream mapping. With time-frequency features, very simple mapping networks (L-vanilla linear network and NL-three-layer MLP) can achieve the SOTA in any circumstance. **The internal mechanisms of FBM variants are provided in Figure 6 in Appendix. Assuming the look-back window  and forecast horizon , FBM-L decomposes the original time series into pieces at different frequency granularities. Thus, our FBM-L mapping uses a “nn.Linear (336  169, 96)”, and the FBM-NL includes two additional nn.Linear layers with activation functions, see Figure 6 in Appendix for details." to the revised Introduction.
>
> 3. Please add your clarification "The Fourier transform is to decompose a time series hierarchically to remove noise signals. For example, for a two-week hourly time series, with day effects and noises only, after Fourier transform, day effects fall in the frequency levels that are multiples of , and noises fall in all the other frequency levels, making mapping much easier. However, the existing mapping in the frequency space completely ignores the time dependent frequency information and faces two inconsistency issues. This is because a Fourier basis function is time-dependent when the input length is not divisible by the frequency level (see Figure 1 in the PDF). " to the revised Introduction

---

> ### Comment · Reviewer_Z6Ax · 2024-08-07
>
> 2. Why only the horizon of 96 and 720 are provided in Table 1 in the attached file? Where are the horizon lengths of 192 and 336? Without a complete experiment result, the analysis will not be sound.

---

> ### Comment · Reviewer_Z6Ax · 2024-08-07
>
> If your comparative experiment is OK (missing horizon length of 192 and 336 now) and you prove that you revise the Introduction section. This manuscript's score can be updated to 6, otherwise it should be 4 for incomplete comparison and introduction.

---

> ### Author Response · Authors · 2024-08-08
> **Extra ablation study and empirical thinking**
>
> We thank you for your further suggestion. We add new results of the horizon of 96 and 720 to Table 1 in the PDF file to address your previous suggestion. Due to the space limit, we could not incorporate all results into that table.
>
> However, the full experiment results of the horizon of 96 to 720 have already been conducted and are presented in Tables 1 and 2 of the paper, respectively. Below, we consolidate them into a single table for better comparison of FBM-L vs. NLinear and FBM-NP vs. PatchTST over the entire horizons, including the results at horizon L = 192 and L = 336 on the eight datasets.
>
> |Model|ETTh1|ETTh1|ETTh2|ETTh2|ETTm1|ETTm1|ETTm2|ETTm2|Electricity|Electricity|Traffic|Traffic|Weather| Weather|Exchange|Exchange|
> |-|-|-|-|-|-|-|-|-|-|-|-|-|-|-|-|-|
> |Length|192|336|192|336|192|336|192|336|192|336|192|336|192|336|192|336|
> | FBM-L-MAE | 0.411| 0.420 |0.374|0.376 | 0.364| 0.384|  0.290|0.326 | 0.216 | 0.232| 0.237 |0.244 |  0.247| 0.285  |0.309 | 0.421|
> |NLinear-MAE|0.426| 0.435| 0.387| 0.395| 0.374| 0.390 | 0.294| 0.331| 0.219| 0.237| 0.245| 0.253| 0.262| 0.296| 0.316| 0.426|
> |FBM-L-MSE|0.403|0.418| 0.333|0.321|0.337|0.371|0.219|0.273| 0.129|0.147| 0.307 |0.314 | 0.203 | 0.252  |  0.195| 0.347 |
> |NLinear-MSE |0.421| 0.435| 0.350 | 0.344|0.347| 0.377| 0.223| 0.277|0.128| 0.148| 0.311| 0.319| 0.220 | 0.265| 0.203| 0.356|
> |FBM-NP-MAE|0.416 |0.438|0.382 |0.411|0.368| 0.389|0.296 |0.331|   0.210 |0.229 | 0.234 | 0.239 |0.245 |0.285  |0.312 |0.425 |
> |PatchTST-MAE| 0.422 | 0.436 | 0.378 | 0.385| 0.370 |0.394 | 0.306 |0.336 | 0.215|0.234 | 0.232 |  0.240  |0.246 | 0.285 | 0.325|0.435 |
> | FBM-NP-MSE |0.407 |0.433 |0.344| 0.374| 0.334| 0.371| 0.224 |0.277 | 0.121 |0.140 | 0.293 |0.298 |0.198 |0.248 |0.196 | 0.353|
> |PatchTST-MSE|0.417 | 0.431 |0.341 | 0.332 |0.333 | 0.363| 0.255| 0.285 |0.123 |0.142 | 0.289 | 0.295  |0.200  | 0.252 |0.210  | 0.366|
>
> Please be noted, all the average MSE and MAE results of FBM-L, NLinear, FBM-NP, and PatchTST reported in the rebuttal are based on full experiments (L=96,192,336,720). We use the same hyperparameters for FBM-NP as the optimal hyperparameters for PatchTST without tuning. Even then, FBM-NP still performs better than PatchTST at most of the time. It is important to note that FBM-L and NLinear don’t have any hyperparameters, as NLinear simply uses the layer nn.Linear(T,L), making it easier to compare. FBM-L always performs better than NLinear in all cases, as shown w.r.t. MAE, which demonstrates the effectiveness of time-frequency features.
>
> **Empirically why FBM works well**
>
> The empirical reason for the effectiveness of the time-frequency feature is that FBM shares the similarity with methods like DLinear and Autoformer, where modeling performance can sometimes be improved using a moving average to separate trend and seasonal effects.  However, this approach is not always effective because it requires determining the appropriate kernel size for the moving average. In our method, we use the Fourier bases to decompose all potential effects into pieces (e.g., 2-hour cycles, 24-hour cycles (daily), 168-hour cycles (weekly), and k-hour cycles), their different effects can fall into different frequency levels. Then, FBM can consider all the potential effects hierarchically, making downstream mapping much easier. Additionally, the Fourier bases includes time-dependent and time-independent basis functions, which further help automatically separate trend and seasonal effects. The mapping allows the model to consider all the potential effects interactively and hierarchically and remove noises in both time and frequency spaces. You can think of FBM as an upgraded version of DLinear, while DLinear only performs better than NLinear in a very small cases, but our FBM-L consistently outperforms NLinear at all the times.
>
> In conclusion, FBM separates all potential effects and allows neural networks to consider those effects automatically and interactively, while DLinear only considers the trend and seasonal effects and whether it is a good separation largely depends on the kernel size you choose.

---

> > ### Comment · Reviewer_Z6Ax · 2024-08-08
> >
> > cool response. Please really add our suggestion to your manuscript. Here is your rate of 6.

---

> > > ### Author Response · Authors · 2024-08-08
> > >
> > > Thank you for your prompt feedback and positive acknowledgement of our responses.
> > >
> > > Definitely the changes will be reflected in the final version. In fact, we have revised the introduction, as illustrated in the various responses and the two last paragraphs shown to you.
> > >
> > > We thank you for referring to our responses to the other reviewers. We'll timely work on the responses to any further questions and suggestions by other reviewers.
> > >
> > > Please be noted that we have responded to your comments following **Reviewer akVH**'s questions on visual assessment and computational complexity, and we are further working on additional update on **akVH**'s new feedback, which mostly has been addressed in the attached PDF file for all reviewers already.
> > >
> > > Do you still have any comments or suggestions? Please kindly let us know, we will continuously try our best to address them and improve the paper quality.

---

> ### Author Response · Authors · 2024-08-08
> **Revised introduction**
>
> Thank you for your further suggestions on adding the revised explanation to the introduction.
>
> Below, we show the last two revised paragraphs of the introduction addressing your comments and suggestions. The final version could be slightly different as more citations will be added and with our continuous improvement.
>
> ***Here is the revised last two paragraphs of Introduction**
>
> Lastly, Fourier-based time series modeling emerges as a new paradigm to remove noise signals by considering diverse effects hierarchically at different frequency levels. However, methods like FEDformer, FreTS, FiLM, FGNet, and FL-Net use real and imaginary parts as inputs to their mapping networks but cannot easily interpret their coefficients because the crucial information is stored in the amplitude and phase of each cycle. This leads to inconsistent starting periods and series lengths, which are often ignored in existing research. Other methods e.g., CrossGNN and TimesNet use the top-k amplitudes to filter noises. However, a higher amplitude does not necessarily indicate useful frequency, and a lower amplitude is not necessarily useless. More importantly, Figure 1 shows that a Fourier basis function is time-dependent when the input length is not divisible by a certain frequency level. Consequently, the mapping in the frequency space is not enough and fails to capture time-frequency relationships.
>
> We provide a new perspective that real and imaginary parts can be interpreted as the coefficients of cosine and sine basis functions at different  granularities of frequencies. However, existing Fourier-based methods do not involve such basis functions, thus failing to interpret these coefficients correctly, as shown in Figure 1. Accordingly, we propose the Fourier Basis Mapping (FBM) by incorporating basis functions to extract more efficient time-frequency features to solve two inconsistency issues, making the downstream mapping much easier. With time-frequency features, very simple mapping networks (L-vanilla linear network and NL-three-layer MLP) can achieve the SOTA in any circumstances. We evaluate our insights through three FBM variants against four categories of LTSF methods: (1) Linear method: NLinear; (2) Transformer-based methods: FEDformer, BasisFormer, iTransformer, Pathformer, and PatchTST; (3) Fourier-based methods: FEDformer, FreTS, N-BEATS, CrossGNN, TimesNet, and FiLM; and (4) MLP-based methods: N-BEATS, FreTS, and TimeMixer. Both FBM-L and FBM-NL achieves the SOTA performance of LTSF on eight real-world datasets or of STSF on datasets PEMS.

---

> ### Author Response · Authors · 2024-08-08
> **More clarification**
>
> Thanks for your suggestions; the clarifications about motivation, internal mechanisms, contributions and performance evaluation, etc. will be reflected in the introduction and evaluation sections respectively.  In addition, a fairer evaluation will be conducted that our FBM variants are listed as independent columns in Table 1.  The details of the revised contents are included in our global reply to all reviewers.

---

### Author Rebuttal · Authors · 2024-08-07

**Please refer to the one-page PDF file attached here for more information.**

We thank all reviewers for constructive comments and suggestions, and expect positive feedback on the unique innovation, extensive experiments, and competitive performance. During the rebuttal, we have addressed all comments and produced a new version of the paper. Major works include:

1.	Clarifying the motivation and workflow of our Fourier Basis Mapping (FBM) model by redrawing Figure 1 (shown in the PDF file) and revising Introduction.
2.	Adding ablation studies for FBM-L vs NLinear and FBM-NP vs PatchTST, with results in Table 1 in the PDF.
3.	Comparing FBM with four extra SOTA methods: iTransformer, TimeMixer, TimesNet, and Pathformer, with results in Table 2.
4.	Testing short time series forecasting (STSF) on the PEMS datasets with results in Table 3.

**Introduction and Figure 1**

First, we revise Introduction to improve readability:

1. para 1 summarizes the significance and challenges of long time-series forecasting (LTSF) and introduces the existing deep LTSF methods.
2. para 2 analyzes the gaps of the Fourier-based method, disclosing two major issues that are ignored: inconsistent starting cycles, and inconsistent series lengths.
3. para 3 proposes our unique insight into addressing the above issues from the basis function perspective, and summarizes the main ideas and contributions of FBM as well as the evaluations of FBM against existing LTSF methods.

Second, we redraw Figure 1 as shown in the PDF to emphasize the distinctive features of FBM against existing methods. Existing Fourier-based methods involve mapping in frequency space, whereas FBM is conducted in both time and frequency space.

Existing methods like FEDformer, FreTS, and FiLM use real and imaginary parts as inputs to their mapping networks but cannot easily interpret their coefficients because the crucial information is stored in the amplitude and phase of each cycle. This leads to inconsistent starting periods and series lengths issue, which are often ignored in existing research. Other methods e.g. CrossGNN, TimesNet use the top-k amplitude to filter noise. However, a higher amplitude does not necessarily indicate useful frequency, and a lower amplitude is not necessarily useless. More importantly, Figure 1 shows that a Fourier basis function is time-dependent when the input length is not divisible by a certain frequency level. Consequently, the mapping in frequency space fails to capture the time-frequency relationships.

By interpreting real and imaginary parts as the coefficients of cosine and sine waves at different frequencies, FBM decomposes an input time series into $\frac{T}{2}+1$ pieces with hierarchical frequency granularity, and T is the sequence length. With time-frequency features, FBM achieves state-of-the-art (SOTA) performance using either a vanilla linear layer or a three-layer MLP on eight real-world datasets for LTSF and PEMS datasets for STSF.

**Table 1: new results on effectiveness**

Table 1 in the PDF shows the results of FBM-L vs. NLinear and FBM-NP vs. PatchTST. FBM-L outperforms NLinear on all datasets and forecast horizons, demonstrating the effectiveness of time-frequency features, as both are linear networks. The average MSE and MAE of NLinear are 0.3135 and 0.3395, respectively, which drops to 0.3034 and 0.3297 for FBM-L. FBM-NP performs better than PatchTST in most of the time, as both are PatchTST networks. For PatchTST, the average MSE and MAE are 0.3079 and 0.3360, respectively, which drop to 0.3058 and 0.3340 for FBM-NL. The improvement on PatchTST is less significant because PatchTST primarily considers the time domain.

**Table 2: new results for more baselines**

Requested by Reviewers akVH and 8ZRG, Table 2 includes results by comparing methods: TimesNet, PathFormer, iTransformer, and TimeMixer on the ETTh1 and ETTh2 datasets. It shows that either FBM-L or FBM-NL consistently achieves the SOTA performance in all circumstances. Additionally, by comparing FBM-NL with TimeMixer, we find that FBM-NL performs better than TimeMixer most of the time (6 out of 8), as both use the MLP network. It is worth mentioning that TimeMixer's structure and hyperparameters are well-tuned for each dataset across different forecast horizons. However, FBM-NL uses the same three-layer MLP and the same hyperparameters for each dataset across different horizons for better reproducibility, see Figure 6 in Appendix.

All the experiment results are conducted with an input sequence length of 336, rather than 96, except for PathFormer, as its hyperparameters do not work with 336. This is consistent with the settings in the paper. A length of 336 always produces better results than 96. Regarding the score reported in the following papers, Group A (Seq len: 336) consistently reports lower MSE and MAE than Group B (Seq len: 96).

Group A:

1.	PatchTST: A Time Series is Worth 64 Words: Long-term Forecasting with Transformers
2.	NLinear: Are Transformers Effective for Time Series Forecasting

Group B:

1.	PathFormer: Multi-scale Transformers with Adaptive Pathways for Time Series Forecasting
2.	iTransformer: Inverted Transformers Are Effective for Time Series Forecasting
3.	TimeMixer: Decomposable Multiscale Mixing for Time Series Forecasting
4.	TimesNet: Temporal 2D-Variation Modeling for General Time Series Analysis

**Table 3: new results on STSF**

Table 3 shows new STSF results on the PEMS datasets, verifying the effectiveness of our FBM. Although FBM primarily addresses the LTSF challenges, it also achieves the best for STSF. In STSF, we change the last hidden state of FBM-NL from 1440 to 720.

We share the LTSF codes in supplementary, allowing to verify our reported experimental results, and will be released on GitHub later with STSF.

**Conclusion**

FBM-L and FBM-NL are listed as two independent columns in Table 1 for a fairer comparison. Four SOTA baseline methods and STSF results are added to the paper.

---

### Decision · Program_Chairs · 2024-09-25

**Decision:**

Accept (poster)

**Comment:**

The paper leverages the Fourier Transform to frequency domain information in predictive models for temporal data. Reviewers Z6Ax and akVH appreciated the novelty of the method, its combination of learnable and non-learnable components, the model selection tailored to the data, and the extensive experiments showing the potential of the method. However, both reviewers indicated lapses in the presentation. Additionally, reviewer Z6Ax recommended comparative studies with the same decoder, while Reviewer akVH requested more comparisons to the state of the art. The authors have provided extensive additional experiments and have greatly improved their introduction and figure. The changes were appreciated by the two reviewers, who agreed to accept the paper. Moreover, reviewer akVH has replicated some of the results reported in the paper (comparison of the model with patchTST) and found that the numbers reported are reliable. Reviewer 8ZRG considered the technical contribution to be incremental and the method not clearly explained, while also pointing out some s.o.t.a models for comparison. The authors have provided an extensive response, which, by my assessment, resolves at least the issues with the comparisons against s.o.t.a., and clarifies the contributions. Notably, Reviewer Z6Ax also pointed Reviewer 8ZRG to a part of the response that appears to clarify the contribution and address a concern. Reviewer 8ZRG has chosen to decrease their score, which is extremely unusual at this stage, and I did not find the justification for it convincing. From their statement "this paper is not ready for being published in NeurIPS", one might gather that the decrease in score is meant to prevent this paper from being published. Reviewer 8ZRG's final set of concerns was also addressed by the authors. It seems that at least some of the results were reproduced by the other reviewer (relevant to point 3), while MAE and MAPE were indeed used by the contender method (point 1), then two of the 3 final points are addressed.
Overall, my assessment is that this paper constitutes a valuable contribution to the field and warrants acceptance. However, the authors should be extremely mindful when preparing the camera-ready and incorporate all the reviewers' suggestions to improve the clarity of the presentation and present the new experiments.